# Function Space Diversity for Uncertainty Prediction via Repulsive Last-Layer Ensembles

## Abstract

Bayesian inference in function space has gained attention due to its robustness against overparameterization in neural networks. However, approximating the infinite-dimensional function space introduces several challenges. In this work, we discuss function space inference via particle optimization and present practical modifications that improve uncertainty estimation and, most importantly, make it applicable to pretrained networks. First, we demonstrate that the input samples, where particle predictions are enforced to be diverse, are critical for model performance. While diversity on training data itself can lead to underfitting, the use of label-destroying data augmentation, or unlabeled out-of-distribution data can improve prediction diversity and uncertainty estimates. Furthermore, we take advantage of the function space formulation, which imposes no restrictions on network parameterization other than sufficient flexibility. Instead of using full deep ensembles to represent particles, we propose a single multi-headed network that shares the backbone across particles. This allows seamless integration to pretrained networks, where this repulsive last-layer ensemble can be used for uncertainty-aware fine-tuning without replicating the full backbone. We achieve competitive uncertainty decomposition, OOD detection, and calibration under distribution shift on synthetic and image-classification benchmarks using ResNet18, ResNet50, and ViT-B/16 backbones.

## 1 Introduction

Deep learning is becoming ubiquitous in our lives, with applications ranging from medical diagnosis to autonomous driving. However, in safety-critical scenarios accurate predictions alone are not sufficient. In addition, models should provide well-calibrated uncertainty estimates to mitigate overconfidence and the potential risks associated with erroneous predictions. Uncertainty methods in deep learning typically distinguish between two types (Kendall & Gal, 2017; Hüllermeier & Waegeman, 2021): (a) Aleatoric uncertainty, which arises from inherent data ambiguity or noise, and (b) epistemic uncertainty, which corresponds to model uncertainty resulting from a lack of knowledge and observations.

Epistemic uncertainty is often estimated by deep ensembles (DEs) (Lakshminarayanan et al., 2017). For the estimate to be accurate, each ensemble member must entail a sufficiently different optimum of the posterior distribution. Particle-optimization variational inference (POVI) (Liu & Wang, 2016; Liu, 2017; Liu et al., 2019) achieves this diversity among ensemble members (i.e., the particles) by incorporating a repulsion term during parameter optimization. However, parameter diversity alone is insufficient. This is because neural networks with different parameters can still represent similar functions. It is thus advisable to avoid this issue by performing inference directly in the function space, enforcing diversity therein (Wang et al., 2019; D'Angelo & Fortuin, 2021). Yet, despite its theoretical appeal, function space (fs) POVI often performs worse than standard DEs in both accuracy and quality of uncertainty estimation (D'Angelo & Fortuin, 2021; Trinh et al., 2024; Yashima et al., 2022). In this work, we discuss the underlying reason for the performance gap; it is not because of the function space diversity but because of the challenges in accurately approximating the infinite-dimensional function space.

It remains practically infeasible to achieve function space diversity over the whole input domain (particularly for high dimensional input data). Therefore, good repulsion samples must not only be *diverse* but also capture the most *relevant parts* of the input domain. The training data itself is generally not rich enough and, as such, insufficient for accurate uncertainty estimation (D'Angelo & Fortuin, 2021; Trinh et al., 2024). We demonstrate how to improve this without sacrificing accuracy: the key is to utilize unlabeled out-of-distribution (OOD) data. If OOD data is unavailable, label-destroying data augmentation[1] can achieve improvements in certain tasks. Our evaluation confirms that well-chosen repulsion samples suppress reliance on spurious features, improve uncertainty estimation, and improve OOD detection across our experiments.

Training and storing multiple ensemble members requires substantial computational resources, especially since the entire ensemble must be optimized jointly to maintain diversity. Function space POVI allows for flexible parameterizations. Drawing inspiration from ensemble distillation (Tran et al., 2020), we propose a multi-headed architecture (see Fig. 1). That is, we first learn a single base network that we subsequently equip with multiple heads – each representing one particle. While such last-layer ensembles (LL-Es) have been proposed in prior work, they are often not diverse enough; thus, we will reintroduce this diversity via repulsion either in parameter or in function space. This last layer repulsion has additional benefits. Specifically, the parameter space has lower dimension and suffers less from overparameterization while the function space allows for explicit prediction differences in relevant input-space regions.

Moreover, last-layer ensembles are especially well-suited for integration with pretrained models, such as vision backbones. Pretraining is typically performed on large, diverse datasets to learn general and expressive features that benefit downstream tasks. This, however, implicitly reduces the applicability of deep ensembles that rely on random initializations to introduce diversity. By equipping a pretrained network with a last-layer ensemble, diversity can still be enforced therein.

From a practical standpoint, this raises important questions: Should one train multiple models from scratch to construct a DE, or is it preferable to use the rich feature set of a pretrained model and achieve diverse predictions through a repulsive last-layer ensemble in function-space (fs-RLL-E)? Setting efficiency consideration aside, can a pretrained model with repulsive heads match (or even surpass) a DE in terms of uncertainty estimation?

**Contributions**

- We propose a parameter-efficient version of POVI via repulsive last-layer ensembles (Section 4.1). Diverse predictions are encouraged by repulsion of the parameters or choosing an appropriate set of repulsion samples for the function space repulsion term (Section 4.2).

- We show that our method can be applied post hoc to pretrained image-classification models, including ResNet18, ResNet50, and ViT-B/16. If the backbone avoids feature collapse, retraining only the repulsive last-layer ensemble is sufficient to obtain meaningful uncertainty estimates (Section 4.3).

- We evaluate the approach on synthetic toy examples and image-classification benchmarks: DirtyMNIST for uncertainty decomposition, CIFAR10-C and CIFAR100-C for calibration under distribution shift, and Food101 and Stanford Cars for transfer learning, using ResNet18, ResNet50, and ViT-B/16 backbones. Across these settings, fs-RLL-E provides competitive uncertainty estimates and OOD detection. On pretrained image-classification models, it often improves OOD detection over unregularized last-layer ensembles and, in some settings, matches or exceeds the uncertainty performance of full deep ensembles (Section 6).

## 2 Background

We consider supervised learning tasks. Let $\mathcal{D} = \{\mathbf{x}_i, \mathbf{y}_i\}_{i=1}^N = (\mathbf{X}, \mathbf{Y})$ denote the training data set consisting of $N$ i.i.d. data samples with inputs $\mathbf{x}_i \in \mathcal{X}$ and targets $\mathbf{y}_i \in \mathcal{Y}$. We define a likelihood model $p(\mathbf{y}|\mathbf{x}, \theta)$ with the mapping $f(\cdot; \theta) : \mathcal{X} \to \mathbb{R}^K$, where $K$ is the number of classes, parameterized by a neural network (NN).

---

[1]Modification of input samples such that the original labels do not apply, e.g., shuffling of random image patches.

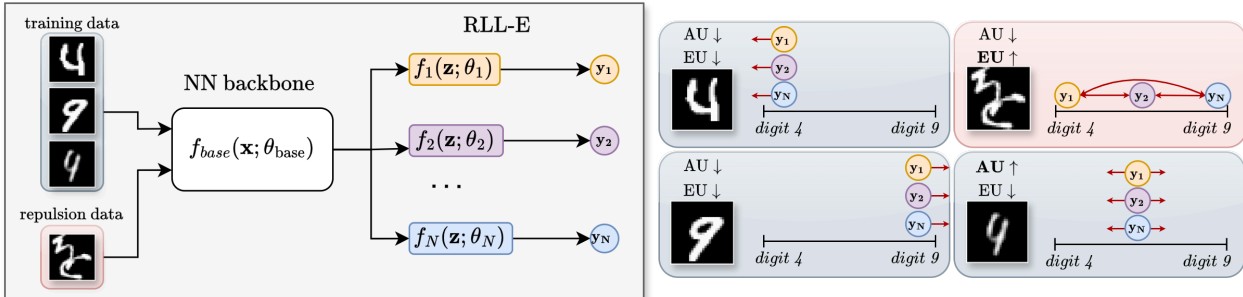

Figure 1: Repulsive last-layer ensemble in function-space (fs-RLL-E), with $N$ particles. Colored dots correspond to particle predictions. Unlabeled OOD data is used as repulsion samples for the function space repulsion loss. Epistemic uncertainty (EU) is the lowest, when all particle predictions agree, and increases with the spread of the particles. Aleatoric uncertainty (AU) rises for ambiguous samples, e.g., the bottom-right digit belonging to both classes.

Bayesian neural networks (BNNs) treat the parameters $\theta$ as random variables rather than deterministic values. A prior $p(\theta)$ is specified, and Bayes' theorem yields the posterior

$$p(\theta|\mathcal{D}) \propto p(\theta)\,p(\mathbf{Y}|\mathbf{X}, \theta).$$

Predictions for a new input $\mathbf{x}_*$ are obtained by marginalizing over $\theta$

$$p(\mathbf{y}_*|\mathbf{x}_*, \mathcal{D}) = \int p(\mathbf{y}_*|\mathbf{x}_*, \theta)\,p(\theta|\mathcal{D})\,d\theta.$$

This integral, however, is generally intractable and motivates approximate inference methods.

**Particle-optimization variational inference** Variational inference approximates the posterior $p(\theta|\mathcal{D})$ by a simpler parametric distribution $q(\theta)$. POVI methods (Liu & Wang, 2016; Chen et al., 2018) aim to provide more flexibility by considering a non-parametric distribution, specified by a discrete set of particles $\{\theta^{(i)}\}_{i=1}^{n}$ according to $q(\theta) \approx \frac{1}{n}\sum_{i=1}^{n}\delta(\theta - \theta^{(i)})$, where $\delta(\cdot)$ is the Dirac function. The particles can then be optimized iteratively via

$$\theta_{l+1}^{(i)} \leftarrow \theta_{l}^{(i)} + \epsilon_l \mathrm{v}(\theta_l^{(i)}),$$

where $\epsilon_l$ is the step size at time step $l$. By viewing the particle optimization as a gradient flow in Wasserstein space, D'Angelo & Fortuin (2021) derive the following update rule that decomposes into an attraction and repulsion term

$$\mathrm{v}(\theta^{(i)}) = \underbrace{\nabla_{\theta^{(i)}} \log p(\theta^{(i)}|\mathcal{D})}_{\text{ATTRACTION}} - \underbrace{\frac{\sum_{j=1}^{n} \nabla_{\theta^{(i)}} k(\theta^{(i)}, \theta^{(j)})}{\sum_{j=1}^{n} k(\theta^{(i)}, \theta^{(j)})}}_{\text{REPULSION}}, \tag{1}$$

where $k(\cdot, \cdot)$ denotes a kernel function. The attraction term drives particles into high-density regions of the posterior distribution, while the repulsion term induces diversity by preventing particles from collapsing into the same optimum. For a single particle, this training procedure reduces to maximum a posteriori (MAP) training; for $n \to \infty$ and a properly defined kernel, it converges to the true posterior distribution (D'Angelo & Fortuin, 2021).

**Why repulsion matters for a finite number of particles** The convergence guarantee to the true posterior distribution $p(\theta|\mathcal{D})$ holds only in the limit of infinitely many particles. Although fascinating from a theoretical perspective, practical importance lies in the analysis of the behavior for a finite number of particles.

Typically, the disagreement between the predictions of ensemble members is used for uncertainty estimation. Following Depeweg et al. (2018), predictive uncertainty can be decomposed into aleatoric and epistemic components, represented as conditional entropy $\mathbb{H}$ and mutual information $\mathbb{I}$:

$$\underbrace{\mathbb{H}\big[\mathbb{E}_{p(\theta|\mathcal{D})}p(\mathbf{y}|\mathbf{x},\theta)\big]}_{\text{Total uncertainty}} = \underbrace{\mathbb{E}_{p(\theta|\mathcal{D})}\mathbb{H}[p(\mathbf{y}|\mathbf{x},\theta)]}_{\text{Aleatoric}} + \underbrace{\mathbb{I}[\mathbf{y};\theta|\mathbf{x},\mathcal{D}]}_{\text{Epistemic}}.$$

Aleatoric uncertainty represents the variability in outcomes due to inherent randomness in the data, while epistemic uncertainty reflects model uncertainty due to limited data. Recent work has raised concerns about the validity of this decomposition (Wimmer et al., 2023; Jiménez et al., 2025). Still, it remains widely used in literature (Kirsch, 2025) and serves as a practical measure for epistemic uncertainty.

For a finite particle approximation, the epistemic term reduces to a Monte Carlo estimate

$$\mathbb{I}[\mathbf{y};\theta|\mathbf{x},\mathcal{D}] \approx \frac{1}{n}\sum_{i=1}^{n} D_{\mathrm{KL}}\left(p(\mathbf{y}|\mathbf{x},\theta^{(i)}) \,\middle\|\, \frac{1}{n}\sum_{j=1}^{n} p(\mathbf{y}|\mathbf{x},\theta^{(j)})\right). \tag{2}$$

If a test sample $\mathbf{x}$ is explained by many disagreeing models $p(\mathbf{y}|\mathbf{x},\theta^{(i)})$ under the posterior $p(\theta|\mathcal{D})$, epistemic uncertainty (EU) is high. Adding training data near $\mathbf{x}$ reduces the space of plausible models and their disagreement.

Given practical constraints on the number of particles (typically five to ten), many posterior modes remain unexplored and the estimate of the epistemic uncertainty is shaped largely by a small number of posterior modes. This limitation stresses the need for guiding particles towards representative and *diverse posterior modes* to avoid underestimation of epistemic uncertainty. Deep ensembles can be viewed as an unregularized case of Eq. (1), lacking a repulsion term. Diversity stems from random weight initialization and often fails to capture truly distinct predictive functions, in some cases assigning lower EU to out-of-distribution inputs than a single model's aleatoric uncertainty (Xia & Bouganis, 2022; Schweighofer et al., 2023b).

POVI methods address this via a repulsion kernel $k\big(\theta^{(i)},\theta^{(j)}\big)$ to avoid mode collapse. In the infinite particle limit, this ensures convergence to the posterior distribution (D'Angelo & Fortuin, 2021; Wild et al., 2023). To improve finite-particle EU estimation, we propose two desiderata:

**D1** *The repulsion term should steer particles towards diverse posterior modes, which provide a useful approximation for the epistemic uncertainty in Eq. (2).*

**D2** *Particles should reach diverse posterior modes from the same initial parameters through the use of the repulsion term. This enables the fine-tuning of pretrained models to better approximate epistemic uncertainty.*

## 3 Where should we enforce diversity?

This leads to the question of where to apply repulsion: parameter space or function space. Unlike parameter-space diversity, function space repulsion acts directly on model predictions.

**Parameter space.** In BNNs, inference is often performed over parameters $\theta$, with repulsion via an $\ell_2$-based kernel (Wang et al., 2019; D'Angelo & Fortuin, 2021):

$$k(\theta^{(i)},\theta^{(j)}) = \exp\left(-\frac{\|\theta^{(i)}-\theta^{(j)}\|_2^2}{\nu^2}\right),$$

where $\nu$ is a hyperparameter, often set by the median heuristic. However, such regularization does not guarantee *diverse predictions* (D1). Overparameterization allows distinct $\theta$ to yield identical outputs due to permutation and scaling symmetries (Pourzanjani et al., 2017). Moreover, in high-dimensional parameter space, Euclidean distances become less informative (Aggarwal et al., 2001), allowing particles to be far apart in $\theta$ yet functionally equivalent.

**Function space.** To target predictive diversity, function space POVI (Wang et al., 2019; D'Angelo & Fortuin, 2021) operates directly on particle predictions $f^{(i)}(\mathcal{X})$, with the repulsion term

$$k(f^{(i)}, f^{(j)}) = \exp\left(-\frac{\|f^{(i)}(\mathcal{X}) - f^{(j)}(\mathcal{X})\|_2^2}{\nu^2}\right),$$

enforcing diverse outputs rather than parameters. Particles are updated via

$$f_{l+1}^{(i)}(\mathcal{X}) \leftarrow f_l^{(i)}(\mathcal{X}) + \epsilon_l \mathrm{v}(f_l^{(i)}(\mathcal{X})),$$

which steers them toward distinct posterior modes with high KL in Eq. (2). However, to solve the problem we must rely on gradient based optimization procedures that in turn require a parameterized representation of the particles. First, each particle $f^{(i)}(\mathcal{X})$ is represented by a specific parameterization $f^{(i)}(\mathcal{X}; \theta^{(i)})$. The parameterization $f^{(i)}(\mathcal{X}; \theta^{(i)})$ must be sufficiently flexible to effectively approximate the underlying function space (Wang et al., 2019).

Moreover, it remains prohibitive to evaluate $f^{(i)}(\mathcal{X}; \theta^{(i)})$ across the entire input domain $\mathcal{X}$. Instead, prior work (Wang et al., 2019) adopted a mini-batch approximation, where the evaluation over the full set $\mathcal{X}$ is replaced with $B$ *repulsion samples* drawn from an arbitrary distribution $\mathbf{x}_{rep} \sim \mu$ with support on $\mathcal{X}^B$. The variational distribution is shown to converge to the true posterior if the posterior can be determined by almost all $B$-dimensional marginals $\{p(f(\mathbf{x})|\mathbf{X}, \mathbf{Y}) : \mathbf{x} \in \mathrm{supp}(\mu)\}$ (Wang et al., 2019).

## 4 Improving function space approximations: Practical choices and implications

Function-space repulsion is conceptually appealing as it mitigates issues of overparameterization and parameter identifiability (Kirsch, 2024) by enforcing diversity directly on the predictions, and thus reducing underestimation of uncertainty. However, empirical gains over unregularized DEs have often been limited, especially on large-scale image tasks (Wang et al., 2019; D'Angelo & Fortuin, 2021; Trinh et al., 2024). We argue that this is largely due to suboptimal practical choices when approximating the function space.

To make this approximation effective, we focus on three design choices: how the particles are parameterized, where diversity is enforced, and how the method behaves in the pretrained setting. Below, we discuss those choices and provide improvements that satisfy our desiderata D1 and D2.

### 4.1 Choice of function parameterization

Particles now represent functions, but gradient-based optimization requires a parameterization $f^{(i)}(\mathcal{X}; \theta^{(i)})$. This choice determines computational cost and representational flexibility.

**Problem.** Prior fs-POVI work (Wang et al., 2019; D'Angelo & Fortuin, 2021) parameterizes each particle as a separate neural network. On large-scale tasks, training and storing multiple full networks is computationally expensive, and parallel training is required due to the repulsion term.

**Our approach.** We propose a *shared base network with multiple prediction heads*:

$$f^{(i)}(\mathbf{x}; \theta_{\mathrm{base}}, \theta_{\mathrm{head}}^{(i)}) = f_{\mathrm{head}}^{(i)}(f_{\mathrm{base}}(\mathbf{x}; \theta_{\mathrm{base}}); \theta_{\mathrm{head}}^{(i)}).$$

The base network $f_{\mathrm{base}}$ provides a shared latent representation; diversity is enforced across the heads $f_{\mathrm{head}}^{(i)}$.

**Justification.** Multi-headed network architectures have been used successfully to distill DEs and replicate their functional behavior (Tran et al., 2020), demonstrating sufficient flexibility of single networks (Hinton et al., 2015). Performing particle optimization in function space mitigates the need for training a full DE prior to distillation. The use of a shared deterministic base network aligns with partially stochastic BNNs, where a subnetwork of the parameters is treated probabilistically. Most prominently, Bayesian last-layer networks are employed as practical means to reduce computational demands (Sharma et al., 2023).

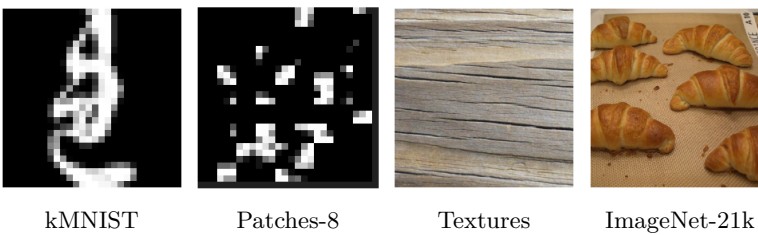

| kMNIST | Patches-8 | Textures | ImageNet-21k |

Figure 2: Examples of repulsion samples.

## 4.2 Choice of repulsion samples

While the shared-base parameterization addresses the computational cost of representing particles, the usefulness of function space diversity also depends on the repulsion samples on which it is enforced. This becomes particularly important when a shared backbone reduces diversity from independent initialization.

**Problem.** Evaluation of the function space repulsion term requires selecting a set of *repulsion samples* $\mathbf{x}_{rep} \in \mathcal{D}_{\mathrm{rep}}$. This choice impacts the valid input domain for uncertainty estimates of fs-POVI methods. Prior work proposed drawing repulsion samples from the kernel density estimate over the training data (Wang et al., 2019), or from the training data directly (D'Angelo & Fortuin, 2021), which ties the BNN approximation to the training distribution and thus does not guarantee reliable uncertainty estimates in OOD settings. In high-dimensional spaces, drawing random samples from the entire input domain is infeasible, requiring restriction to an informative subset that covers the domain of interest.

**Our approach.** We use *unlabeled OOD data* when available, or apply label-destroying augmentations to the training data. In image classification, this often includes natural images from varying distributions. We can thus exploit the abundance of available unlabeled image data. For example, using kMNIST as repulsion samples for models trained on MNIST, or Textures for models trained on CIFAR10/100, leverages natural variability across different image sets. If unlabeled OOD data is unavailable, repulsion samples can be generated from the training data by label-destroying data augmentation techniques. One such method is the random shuffling of image patches to destroy the shape information of objects that is crucial for human perception (see Fig. 2). The effectiveness of repulsion samples depends on their informativeness and domain coverage. If the samples are too close to the training distribution, they may fail to induce meaningful diversity and can degrade in-domain accuracy. If they are too far from the data manifold, such as random noise, they may have little or no effect on the model's behavior. Selecting effective repulsion samples therefore requires some knowledge of what constitutes in-distribution (ID) and out-of-distribution inputs. This knowledge enables us to guide diversity in function space in a more targeted way, compensating for the lack of random weight initialization.

**Justification.** Enforcing diversity directly on the training data has been shown to degrade classification accuracy by artificially inflating epistemic uncertainty at data points where independent training would yield confident predictions Abe et al. (2022); Jeffares et al. (2023). This approach often fails to detect OOD data, which may be characterized by spurious features present in the training data or features that are completely absent in the training set. Using unlabeled OOD data as repulsion samples provides an effective solution to these challenges. These samples may contain features that are spurious or absent in the training data, allowing the models to meaningfully enforce diversity and improve OOD detection capabilities. Similarly, label-destroying data augmentation mitigates robust features that are indicative of the class label. Compared to methods that rely on feature density to detect OOD data, repulsion samples offer the benefit of learning to ignore spurious features that may be present in the training data.

## 4.3 Post-hoc uncertainties for pretrained models

There are many scenarios where using a pretrained model is desirable. We may wish to equip an existing model with post-hoc uncertainty estimates without retraining from scratch, or finetune a base model pretrained on large datasets for a smaller target dataset. Pretraining yields strong, generalizable features,

improves data efficiency, and speeds convergence. While some attribute these gains mainly to faster optimization (He et al., 2019), others find improved calibration even for single models (Hendrycks et al., 2019a). However, when the objective is to capture epistemic uncertainty, pretraining counteracts the benefit of deep ensembles that rely on random initializations with independent training to achieve diversity.

In such cases, multi-headed networks provide a principled way to estimate post-hoc uncertainties for the pretrained base. While LL-E approaches have been used before (Schweighofer et al., 2023b), they often lack diversity due to the shared base model. The repulsion term allows us to reintroduce this diversity and decouple representation learning (via the base model) from uncertainty estimation (via the diverse last-layer ensemble). If OOD-relevant features are mapped to ID features in feature space, known as feature collapse (van Amersfoort et al., 2020), the ensemble heads may not achieve the desired diverse predictions. Fortunately, many modern architectures include mechanisms that mitigate collapse.

**Spectral normalization and residual connections.** Distance-aware feature representations can be encouraged by imposing bi-Lipschitz constraints $K_L d_I(\mathbf{x}_1, \mathbf{x}_2) \leq d_F(f_{\text{base}}(\mathbf{x}_1), f_{\text{base}}(\mathbf{x}_2)) \leq K_U d_I(\mathbf{x}_1, \mathbf{x}_2)$, where $d_I$ and $d_F$ denote distances in input and feature space, and $K_L$, $K_U$ are lower and upper Lipschitz constants. These constraints bound how input distances translate into feature distances. Spectral normalization and residual connections can impose such bounds (Miyato et al., 2018). While most pretrained models are not trained with spectral normalization, residual architectures alone often preserve distance-awareness.

**Pretraining.** Large-scale pretraining, whether supervised, self-supervised, or contrastive (Krizhevsky et al., 2012), produces features that generalize well (Hendrycks et al., 2019a) and promote semantic separation in feature space. This mitigates collapse, but OOD inputs may still activate spurious features learned during pretraining, leading to overconfident errors. Repulsion in function space can approach this by encouraging prediction-head disagreement on such inputs.

## 5 Related Work

**(Repulsive) Deep Ensembles.** Deep ensembles (Lakshminarayanan et al., 2017) often outperform BNNs in accuracy, calibration, and OOD detection (Gustafsson et al., 2020; Ovadia et al., 2019). Repulsive variants promote diversity via kernelized losses (Wang et al., 2019; D'Angelo & Fortuin, 2021), feature/gradient differences (Yashima et al., 2022; Trinh et al., 2024), or disagreement on unlabeled data (Pagliardini et al., 2023). Weight-distribution entropy maximization (de Mathelin et al., 2025) also has improved epistemic uncertainty.

**Function-space inference.** A number of inference methods for BNNs consider the shift from inference in the space of network parameters to the function space Sun et al. (2019); Ma et al. (2019); Burt et al. (2020); Wang et al. (2019); Ma & Hernández-Lobato (2021); Rudner et al. (2022). This allows to specify meaningful prior distributions over the network parameters. Recent work proposes tractable VI via local linearization (Rudner et al., 2022; 2023). POVI methods approximate the posterior distribution using a set of discrete particles to capture its multimodal structure Wang et al. (2019); D'Angelo & Fortuin (2021).

**Auxiliary out-of-distribution data.** Function space inference methods enforce the function prior on a set of input points, in some work referred to as measurement (Sun et al., 2019; Wang et al., 2019; Ma & Hernández-Lobato, 2021) or context samples (Rudner et al., 2022; 2023). In low-dimensional problems, such samples can be obtained by drawing from a distribution with support over the domain of interest (Sun et al., 2019; Wang et al., 2019; Ma & Hernández-Lobato, 2021). For high-dimensional problems with structured data, such as natural images, samples from an OOD data set have shown improvements (Rudner et al., 2022; 2023). Related work on OOD detection methods for single networks has used auxiliary OOD datasets to maximize softmax entropy (Hendrycks et al., 2019b).

**Multi-headed architectures.** Multi-headed networks reduce memory requirements by sharing a backbone (Song & Chai, 2018; Sercu et al., 2016; Lee et al., 2015). Among the first, last-layer ensembles were analyzed by Lee et al. (2015) in terms of accuracy, parameters sharing, and diversity by random parameters versus bagging the data. They have been applied in RL (Osband et al., 2016), distillation (Zhu et al., 2018; Tran et al., 2020), and uncertainty estimation (Valdenegro-Toro, 2023), with diversity encouraged via decorrelation (Zhang et al., 2020; Lee et al., 2022).

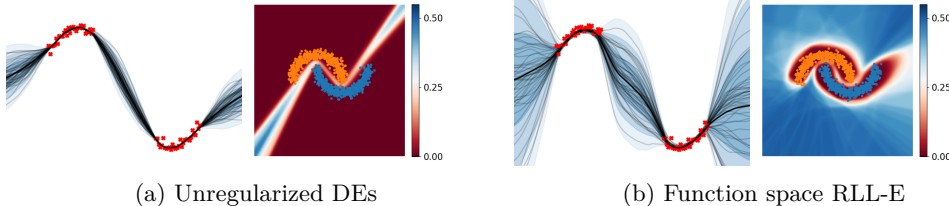

(a) Unregularized DEs           (b) Function space RLL-E

Figure 3: For regression, we show the prediction of individual particles/ensemble members, the mean and the standard deviation. For classification, we show the standard deviation of $p(\mathbf{y}|\mathbf{x}, \theta)$. DEs are highly confident in regions distant from training data (low standard deviation is colored in red), while fs-RLL-E predictions are enforced to be diverse outside of the training data domain.

**Distance-based uncertainty methods.** These relate epistemic uncertainty to distance from training support, typically in latent space (Charpentier et al., 2020; Postels et al., 2020; Mukhoti et al., 2023; Winkens et al., 2020; Liu et al., 2020; van Amersfoort et al., 2020). They require feature regularization (van Amersfoort et al., 2021) via gradient penalties (Gulrajani et al., 2017) or spectral normalization (Miyato et al., 2018). While strong for OOD detection, they often miscalibrate under shift (Postels et al., 2022); our approach benefits from similar regularization but enforces diversity directly in function space.

## 6 Experiments

We evaluate our repulsive last-layer ensembles in two settings: uncertainty for models trained from scratch and post-hoc uncertainty for ImageNet-1k pretrained backbones. In both settings, we test whether diversity lost through backbone sharing can be recovered by repulsion in parameter space or function space.

We begin with synthetic toy examples to illustrate the behavior of our multi-head model and its function space diversity. We then evaluate DirtyMNIST for uncertainty decomposition, CIFAR10-C and CIFAR100-C for calibration under distribution shift, and Food101 and Stanford Cars for transfer learning with pretrained backbones. In the appendix, we provide implementation details (Sec. A), quantify computational overhead (Sec. A.1), describe the datasets (Sec. A.2), further evaluate active learning and OOD detection (Sec. B.1 and B.3), and perform ablations on repulsion sample selection (Sec. B.2).

**Models and baselines.** For the training-from-scratch experiments we use ResNet18 and ResNet50 with spectral normalization; pretrained experiments additionally include ViT-B/16. Baselines comprise a single MAP-trained network, deep ensembles with five members (DE-5), deterministic uncertainty (DDU, Mukhoti et al. (2023)), last-layer Laplace approximation (LL-Laplace, Kristiadi et al. (2020)), and non-repulsive last-layer ensembles (LL-E). Our variants apply repulsion in parameter space (RLL-E) or function space (fs-RLL-E), using identical backbones for fair comparison.

**Metrics.** We report accuracy and negative log-likelihood (NLL), expected calibration error (ECE), and OOD AUROC/AUPR. We use mutual information (Equation 2) for ensemble-based methods, maximum softmax probability (MSP)/ maximum logit/ entropy for the single base model, and GMM density for DDU. Accuracy and NLL are treated as constraints: improvements in uncertainty should not come at the cost of substantially reduced in-distribution performance.

### 6.1 Synthetic data

On two toy examples, we illustrate the effectiveness of the multi-head architecture as a lightweight parameterization and inference in function space. We estimate the epistemic uncertainty for a one-dimensional regression and a two-dimensional classification problem using full DEs with 30 ensemble members and fs-RLL-E. A feed-forward neural network with 3 hidden layers and 128 neurons is used as the base model. The repulsive head consists of 30 particles with a linear layer. Results are shown in Figure 3. Deep ensemble predictions show low uncertainty far from the training data. By performing particle inference in function space, we can enforce diverse predictions outside the training distribution even with a simpler network structure.

### 6.2 Disentangling aleatoric and epistemic uncertainty

**Setup** We first test our method for disentangling aleatoric and epistemic uncertainty on DirtyMNIST, a dataset with ambiguous labels. DirtyMNIST contains 60k clean MNIST digits and 60k ambiguous digits with multiple valid labels (high aleatoric uncertainty). Our repulsive last-layer ensemble aims to model epistemic uncertainty by learning multiple hypotheses that agree on training data but disagree elsewhere. As epistemic uncertainty is difficult to evaluate directly, we use OOD detection as a proxy, following the common assumption that epistemic uncertainty should increase on OOD inputs (de Mathelin et al., 2025).

We evaluate: (i) classification and calibration on DirtyMNIST (Acc., NLL, ECE); (ii) OOD detection between clean MNIST and eMNIST, FashionMNIST, Omniglot, where aleatoric uncertainty often suffices; and (iii) OOD detection between ambiguous MNIST and the same datasets, requiring epistemic uncertainty. All post-hoc methods (DDU, LL-Laplace, LL-E, RLL-E, fs-RLL-E) use the feature space of a ResNet18 MAP model. For LL-E variants, we reinitialize the last linear layer with 10 heads, freeze the backbone, and train with AdamW (LR=0.001). Repulsion is applied in parameter space (RLL-E) or function space (fs-RLL-E).

**Results** Table 1 summarizes the results across all tasks. First, we evaluate in-distribution performance on DirtyMNIST (Accuracy, NLL, ECE). The LL-E and our repulsive variants (RLL-E, fs-RLL-E) mostly improve upon the single-model baseline (MAP). However, the main advantage of our approach becomes evident in the OOD detection tasks. The AUROC and AUPR scores reflect how well the specified uncertainty estimator separates ID and OOD samples across all thresholds. Ideally, epistemic uncertainty should be small on ID samples and large on unseen OOD data.

On the clean-vs-OOD task, even using the aleatoric uncertainty (Entropy) of the single-model (MAP) as OOD detection scores performs reasonably well with an AUROC score of 97.57%. Nonetheless, our RLL-E extension improves detection further (98.63%) and even outperforms the full DE-5 (97.7%), despite using a shared backbone rather than training separate models. The more challenging ambiguous-vs-OOD task reveals the limitations of the single-model baseline. Here, the aleatoric uncertainty detector degrades to AUROC values of 73.76%. Our RLL-E still achieves 93.32%, outperforming the LL-Laplace and the full DE-5. Repulsion in function space using samples from the training distribution itself (DirtyMNIST) leads to degraded OOD detection. In contrast, using kMNIST or training data with label-destroying augmentations (such as Patches-8) introduces beneficial diversity. This results in improved separation of ambiguous and OOD samples, with AUROC values increasing to 95.92% and 96.33%, respectively.

DDU performs best on OOD detection (99.38%), but follows a different strategy by fitting class-conditional Gaussians in feature space. In high-dimensional feature spaces with many classes, the required covariance matrices become memory-intensive. We therefore also compare against a diagonalized version of DDU, which matches the performance of our fs-RLL-E with 96.47%. We revisit this limitation in Section 6.4 when evaluating pretrained backbones.

### 6.3 Covariate shift calibration

**Setup** We next evaluate whether the uncertainty estimates remain reliable under distribution shifts that corrupt the input while preserving the underlying class labels. We evaluate how uncertainty estimators behave under covariate shift on the CIFAR10-C and CIFAR100-C datasets. These datasets apply 19 types of common corruptions (e.g., blur, noise, compression), each at 5 severity levels, to simulate real-world data shifts. We report uncertainty calibration in terms of NLL and ECE, averaged over all corruption types.

**Results** Figure 4 summarizes the calibration performance under corruption. Retraining the last layer without regularization (LL-E) already improves the calibration of the single model (MAP) for both CIFAR10-C and CIFAR100-C. On CIFAR10-C, our function space repulsion (fs-RLL-E) yields further improvements, outperforming LL-Laplace, and achieving calibration competitive with DE-5, while requiring fewer parameters. The improvement can be attributed to better feature selection in the presence of corruptions. If the final classification layer relies on spurious or fragile features, corrupted inputs may trigger high-confidence but incorrect predictions. By enforcing disagreement on repulsion samples, our method promotes diversity in predictive behavior and reduces over-reliance on non-robust features, resulting in better calibration.

Table 1: **Uncertainty decomposition on DirtyMNIST.** Accuracy, NLL, and ECE are reported for the test split of DirtyMNIST. For distinguishing clean, ambiguous and OOD samples we report the AUROC scores. Results are averaged over 5 seeds. Best results are **bold**, second-best are underlined.

| Method | Uncertainty | DirtyMNIST | | | Clean MNIST vs OOD | Ambig. MNIST vs OOD |
| | | Acc. [%] ↑ | NLL ↓ | ECE [%] ↓ | AUROC /AUPR [%] ↑ | AUROC /AUPR [%] ↑ |
|---|---|---|---|---|---|---|
| MAP | MSP | | | | 97.35 / 96.47 | 70.01 / 46.04 |
| | Max Logit | 80.78 | 0.5591 | 2.09 | 98.58 / 98.68 | 73.75 / 52.88 |
| | Entropy | | | | 97.57 / 96.82 | 73.76 / 52.08 |
| DDU | GMM density | 80.78 | 0.5591 | 2.09 | **99.15** / **99.80** | **99.38** / **99.61** |
| DDU (diag) | GMM density | | | | 97.35 / 98.98 | 96.47 / 96.25 |
| LL-Laplace | MI | 80.75 | 0.5612 | 1.97 | 97.94 / 99.31 | 89.83 / 88.94 |
| LL-E | MI | 83.51 | 0.4838 | 1.36 | 98.46 / 99.66 | 92.01 / 94.79 |
| RLL-E (ours) | MI | 83.51 | 0.4839 | 1.37 | 98.63 / 99.37 | 93.32 / 93.82 |
| fs-RLL-E (ours) | | | | | | |
| + DIRTYMNIST | MI | **83.54** | **0.4834** | 1.31 | 96.74 / 97.41 | 48.44 / 29.41 |
| + KMNIST | MI | 82.97 | 0.4961 | **0.92** | 97.76 / 99.55 | 95.92 / 97.25 |
| + PATCHES-8 | MI | 83.04 | 0.4932 | 1.20 | 97.42 / 99.44 | 96.33 / 97.56 |
| DE-5 | MI | 83.32 | 0.5024 | 5.20 | 97.70 / 98.43 | 88.80 / 79.07 |

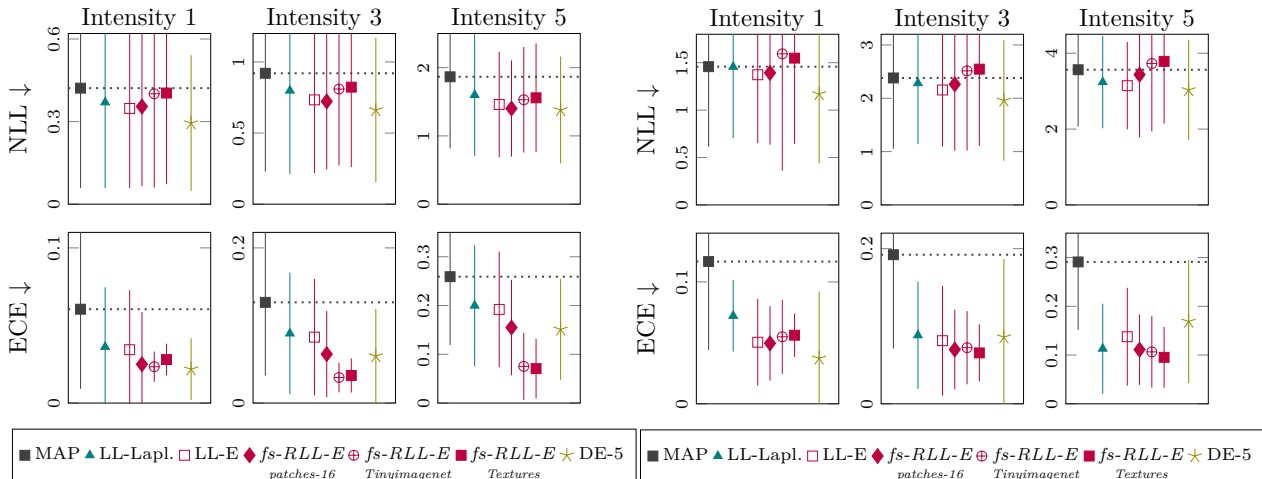

Figure 4: NLL and uncertainty calibration of the different methods on CIFAR10-C (left) and CIFAR100-C (right), for different levels of corruption intensity (columns), averaged over all corruption types. By retraining the last linear layer only, our method fs-RLL-E improves NLL, and achieves similar ECE scores as DE-5.

## 6.4 Transfer learning with pretrained models

**Setup** Finally, we test post-hoc uncertainty estimation on pretrained vision backbones for transfer learning. We study transfer learning on Food101 and Stanford Cars with ImageNet-1k pretrained ResNet18, ResNet50, and ViT-B/16 backbones. Since this setting focuses on post-hoc uncertainty estimation for pretrained backbones, we compare primarily against post-hoc baselines rather than full deep ensembles trained from scratch. Specifically, we compare against LL-E, LL-Laplace, and DDU as uncertainty baselines. For fine-tuning, we jointly optimize the backbone and ensemble head with learning rates of $10^{-5}$ and $10^{-3}$, respectively. For function space repulsion, we use Textures and ImageNet-21k and evaluate OOD detection on SVHN, Places365, ImageNet-O, Country211, and TinyImageNet.

**Results** Table 2 and 3 summarize the ID performance and OOD detection, respectively. We do not primarily expect the repulsion terms to improve in-distribution accuracy or NLL. Instead, their role is to

Table 2: **In-distribution performance (Accuracy [%] / NLL).** We compare base models that are *fine-tuned from ImageNet-1k pretrained weights.* Results are averaged over 10 seeds.

| Method | Food101 | | | Stanford Cars | | |
|---|---|---|---|---|---|---|
| | **ResNet18** (Pretrained) | **ResNet50** (Pretrained) | **ViT-B/16** (Pretrained) | **ResNet18** (Pretrained) | **ResNet50** (Pretrained) | **ViT-B/16** (Pretrained) |
| MAP | 75.72 / 0.94 | 82.43 / 0.72 | 83.14 / 0.78 | 66.57 / 1.25 | 71.68 / 1.07 | 82.07 / 0.66 |
| LL-Laplace | 75.51 / 0.92 | 81.23 / 0.81 | 82.75 / 0.69 | 63.36 / 1.68 | 58.45 / 2.27 | 73.91 / 1.72 |
| DDU (diag) | 75.72 / 0.94 | 82.42 / 0.72 | 83.14 / 0.78 | 66.55 / 1.25 | 71.69 / 1.07 | 82.07 / 0.66 |
| LL-E | 75.82 / 0.94 | 82.37 / 0.72 | 82.89 / 0.78 | 68.20 / 1.17 | 72.00 / 1.06 | 82.70 / 0.64 |
| RLL-E *(ours)* | 75.97 / 0.93 | 82.37 / 0.71 | 82.64 / 0.79 | 68.31 / 1.17 | 72.18 / 1.05 | 82.76 / 0.64 |
| fs-RLL-E *(ours)* | | | | | | |
| + Textures | 75.87 / 0.93 | 82.41 / 0.72 | 83.50 / 0.77 | 68.18 / 1.18 | 71.84 / 1.06 | 82.53 / 0.64 |
| + ImageNet-21k | 75.82 / 0.94 | 82.57 / 0.72 | 83.64 / 0.76 | 68.16 / 1.17 | 71.81 / 1.06 | 82.66 / 0.64 |

Table 3: **Semantic shift detection (AUROC [%] / AUPR [%]).** We compare base models that are *fine-tuned from ImageNet-1k pretrained weights.* Results are averaged over 10 seeds. Best results for each base model are **bold**, second-best are underlined.

| Method | Uncertainty | Food101 | | | Stanford Cars | | |
|---|---|---|---|---|---|---|---|
| | | **ResNet18** (Pretrained) | **ResNet50** (Pretrained) | **ViT-B/16** (Pretrained) | **ResNet18** (Pretrained) | **ResNet50** (Pretrained) | **ViT-B/16** (Pretrained) |
| MAP | MSP | 93.66 / 87.16 | 90.00 / 75.90 | 94.91 / 88.39 | 86.59 / 87.54 | 93.13 / 93.97 | 98.86 / 99.09 |
| MAP | Max Logit | 71.39 / 37.67 | 39.31 / 22.51 | 99.19 / 98.35 | 68.15 / 63.92 | 13.08 / 37.34 | 99.56 / 99.66 |
| MAP | Entropy | 95.62 / 91.75 | 92.63 / 82.76 | 97.13 / 94.24 | 90.28 / 91.01 | 95.92 / 96.59 | 99.62 / 99.73 |
| LL-Laplace | MI | 86.88 / 73.21 | **99.03** / 96.45 | **99.37** / 97.83 | 94.95 / 96.22 | 96.80 / 95.73 | 99.39 / 99.23 |
| DDU (diag) | GMM density | 49.72 / 35.13 | 86.27 / 76.14 | 98.36 / 95.61 | 82.43 / 85.62 | 77.91 / 79.34 | 99.88 / **99.94** |
| LL-E | MI | 87.38 / 65.30 | 96.36 / 92.38 | 98.43 / 97.16 | 93.06 / 93.63 | 96.40 / 96.59 | 99.72 / 99.80 |
| RLL-E *(ours)* | MI | 90.73 / 71.08 | 97.37 / 93.41 | 98.66 / 97.33 | 93.65 / 94.07 | 98.95 / 99.01 | 99.81 / 99.86 |
| fs-RLL-E *(ours)* | | | | | | | |
| + Textures | MI | 95.30 / 89.17 | 97.75 / 94.99 | 98.95 / 97.90 | 99.27 / 99.46 | 98.86 / 99.11 | 99.89 / 99.92 |
| + ImageNet-21k | MI | **97.31** / **94.70** | 98.73 / **97.48** | 99.15 / **98.44** | **99.35** / **99.54** | **99.36** / **99.50** | **99.89** / 99.91 |

improve epistemic uncertainty while preserving predictive performance. In practice, the repulsive variants keep in-distribution metrics close to the corresponding single-model and LL-E baselines.

Clear benefits emerge in the OOD detection tasks (Table 3), especially for ResNet18. Relative to LL-E, fs-RLL-E with Textures improves Food101 from 87.38% / 65.30% to 95.30% / 89.17% in AUROC / AUPR and Stanford Cars from 93.06% / 93.63% to 99.27% / 99.46%. Using the more diverse ImageNet-21k repulsion dataset further improves or matches these OOD results across the pretrained backbones we consider.

For stronger backbones such as ResNet50 and ViT-B/16, the gains remain positive but smaller, indicating that better representations already provide stronger uncertainty estimates. LL-Laplace is strongest on Food101 with ResNet50 and ViT-B/16, though with slightly weaker ID performance. DDU is competitive on ViT-B/16 but less reliable on convolutional models, likely because of its class-conditional Gaussian assumptions and diagonal covariance approximation. Overall, LL-Es improve over the single-model baseline, and RLL-E / fs-RLL-E further improve uncertainty estimation. The main effect is on uncertainty quality, while ID accuracy and NLL remain similar. Using structured auxiliary OOD samples for repulsion consistently improves OOD detection over both LL-E and RLL-E.

Appendix B.2 analyzes repulsion sample choice more systematically by comparing ID samples, structured auxiliary OOD datasets, and mixtures interpolating auxiliary OOD samples toward ID images or random noise. These results show that structured auxiliary OOD samples are the most reliable repulsion data in our experiments, whereas samples from the training data and unstructured noise are considerably less effective.

**Practitioner's guide**   The choice between methods mainly depends on two practical constraints: whether full backbone replication is affordable and whether informative repulsion samples are available. If compute is not limiting, deep ensembles remain a strong baseline, but their cost scales with the number of independently trained models. When this is prohibitive, LL-E provides a low-overhead alternative by sharing the backbone and adding only multiple prediction heads. This improves efficiency, but may lack useful diversity.

RLL-E encourages such diversity without replicating the full network. RLL-E is suitable when no informative auxiliary repulsion samples are available, since it enforces diversity between the parameters of the prediction heads. fs-RLL-E is most useful for OOD detection or epistemic uncertainty under shift, where repulsion samples can guide diversity toward relevant input regions. In our transfer-learning experiments, structured auxiliary OOD data works best, while ID samples and pure noise are less reliable.

LL-Laplace can be a competitive post-hoc baseline when suitable repulsion samples are difficult to choose. DDU is attractive when feature-space density modeling is well behaved, but depends on how well its Gaussian assumptions fit the learned representation. Thus, LL-E and RLL-E are most useful when efficiency is central, whereas fs-RLL-E is most appropriate when uncertainty under shift is the target and informative auxiliary samples are available. In Appendix A.1, we provide an analysis of the computational trade-offs between MAP, LL-E, RLL-E, fs-RLL-E, and DEs, covering parameter count, memory requirements, training overhead, and inference cost.

## 7   Conclusion

We have shown that particle optimization in function space is not limited to DE architectures. By representing particles as prediction heads on top of a shared base model, function space diversity can be introduced without replicating the full network. This provides a principled way to equip pretrained vision models with post-hoc uncertainty estimates and to incorporate knowledge about where predictive diversity should be encouraged. Across synthetic and image-classification benchmarks, shared-backbone repulsive ensembles improve uncertainty estimates while reducing parameter count, inference, and training cost relative to full deep ensembles.

Useful function space diversity depends strongly on where the repulsion objective is evaluated: Structured auxiliary data or suitable label-destroying augmentations can improve OOD detection, whereas repulsion on training data alone or unstructured noise can be harmful or ineffective. Overall, our results suggest that an RLL-E is most useful when a strong shared representation is available and diversity can be encouraged on informative input regions. In this setting, fs-RLL-E can improve uncertainty estimates while avoiding replication of the full backbone.

**Limitations and future work**

A central limitation of function space repulsion is the requirement to select input samples on which predictive diversity is enforced. If those repulsion samples do not cover the domain of interest during deployment of the model, the function space repulsion term may fail to improve the quality of epistemic uncertainty estimates and their effectiveness in downstream tasks such as OOD detection. While our experiments provide practical guidance for vision classification, more rigorous criteria for selecting repulsion samples and understanding their implications for uncertainty estimates remain important directions for future work. We further plan to explore data augmentation schemes for the generation of repulsion samples for different tasks and their respective limitations.

Complementary to this, mutual-information-based epistemic uncertainty has been criticized as an incomplete decomposition of uncertainty (Wimmer et al., 2023; Schweighofer et al., 2023a; Jiménez et al., 2025). While we use mutual information operationally as an ensemble-disagreement measure, future work should investigate how different uncertainty decomposition schemes influence the resulting epistemic uncertainty estimates.

Lastly, our empirical evaluation is currently limited to vision classification tasks. In future work, we want to extend our approach to real-world regression tasks and other data modalities.

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

# A    Experimental details

For particle-based inference in function space (Wang et al., 2019; D'Angelo & Fortuin, 2021), we relied on the implementation available at `https://github.com/ratschlab/repulsive_ensembles`, and for DDU (Mukhoti et al., 2023) at `https://github.com/omegafragger/DDU`. Table 4 summarizes relevant hyperparameters for training the base networks and our repulsive last-layer ensemble (RLL-E). For networks with spectral normalization we follow the implementation of (Mukhoti et al., 2023). Online spectral normalization with a one step power iteration is applied to convolutional weights, and exact spectral normalization is applied to 1x1 convolutional layers (Mukhoti et al., 2023). For the optimization of the ensemble head, we use a batch size of 32 for the training data and 16 for the repulsion samples for all image classification tasks.

We additionally compare two post-hoc uncertainty baselines built on top of the same trained single model. For DDU (Mukhoti et al., 2023), we fit class-conditional diagonal Gaussian densities to penultimate-layer ID features and use the resulting feature-space density for OOD detection. For LL-Laplace (Kristiadi et al., 2020), we place a Laplace approximation over the last-layer weights using a Kronecker-factored Hessian approximation and perform prediction with 25 Monte Carlo neural-network predictive samples; the resulting posterior predictive distribution is used for both ID metrics and OOD uncertainty scores.

Table 4: Implementation details and hyperparameter for the different experiments.

| Task | Architecture | Hyperparameter | Value |
|---|---|---|---|
| Image classification base network | ResNet-18 ResNet50 | Epochs | 50 (DirtyMNIST) 300 (CIFAR10/100) |
| | | Optimizer | SGD |
| | | Learning rate | 0.1 0.01 – epoch 25 (DirtyMNIST), epoch 150 (CIFAR10/100) 0.001 – epoch 40 (DirtyMNIST), epoch 250 (CIFAR10/100) |
| | | Momentum | 0.9 |
| Active learning base network | ResNet-18 | Epochs | 20 |
| | | Optimizer | Adam |
| | | Learning rate | 0.001 |

## A.1    Computational overhead

For the image-classification experiments, the main efficiency benefit of LL-E, RLL-E, and fs-RLL-E is backbone sharing. Instead of replicating the full network, we reuse a single backbone and add an ensemble head consisting of linear layers only. The additional parameter count is therefore determined by the feature-space dimension $d$, the number of classes $K$, and the number of particles $n$, i.e., $(d \times K + K) \times n$. Table 5 summarizes the feature dimensions of the backbone architectures used in our experiments and therefore determines the size of the additional ensemble head.

At inference time, the repulsion losses are not evaluated. The backbone features are computed once and predictions are obtained by applying $n$ lightweight classifier heads. Consequently, the inference cost of LL-E, RLL-E, and fs-RLL-E remains close to that of a single model. In contrast, a deep ensemble requires $n$ separate backbone evaluations. During training, LL-E only incurs the cost of evaluating the additional heads, RLL-E additionally evaluates the parameter-space repulsion term, and fs-RLL-E introduces the largest overhead through an additional repulsion batch and pairwise prediction distances between heads. Memory consumption is similarly dominated by the repulsion batch and can be reduced through gradient accumulation at the cost of increased training time.

To quantify these trade-offs, we profile a single ResNet18 / ResNet50, LL-E, RLL-E, fs-RLL-E, independently trained DE-10, and jointly trained RDE-3 on the same NVIDIA RTX A4000. All last-layer ensemble methods use $n = 10$, and the profiling benchmark uses effective in-distribution and repulsion batch sizes of 16.

Tables 6 and 7 summarize the relative parameter counts, inference overheads, training overheads, and peak memory usage for ResNet18 and ResNet50, respectively. Several observations are noteworthy. First, the parameter overhead of the shared-backbone methods is small: increasing the ensemble size from one model to ten members only increases the parameter count from 11.23M to 11.69M for ResNet18 and from 23.71M to 25.58M for ResNet50. In contrast, DE-10 scales almost linearly with the number of ensemble members. Second, inference overhead remains negligible for all shared-backbone methods (approximately $1.01\times$–$1.02\times$ a single model), whereas DE-10 requires roughly ten times the inference cost and RDE-3 approximately three times the inference cost. This reflects the fact that the backbone is evaluated only once in the shared-backbone setting.

The main computational cost of function space repulsion appears during training. For ResNet18, fs-RLL-E increases training cost from $1.00\times$ to $2.07\times$, while DE-10 increases training cost to $9.84\times$. Similar trends are observed for ResNet50. Thus, function space repulsion is not free, but its cost remains substantially below that of training multiple independent backbones. Jointly trained repulsive deep ensembles (RDE-3) highlight this trade-off further: despite using only three ensemble members, they require substantially more memory and training time because all backbone activations must be stored simultaneously.

Table 8 reports absolute latency measurements. These results confirm that the relative overheads correspond to small absolute differences for the shared-backbone methods. For example, on ResNet50, inference increases only from 1.276 ms to 1.306 ms per image when moving from a single model to fs-RLL-E, whereas DE-10 increases latency to 12.774 ms. This supports our claim that the proposed methods preserve near-single-model inference behavior.

Finally, Table 9 illustrates the memory-runtime trade-off of gradient accumulation for fs-RLL-E. Reducing the repulsion microbatch size[2] substantially lowers peak memory consumption, especially for larger backbones such as ResNet50, but increases training time because the same effective batch must be processed through multiple forward and backward passes. This provides a practical mechanism for applying function space repulsion under limited GPU memory budgets.

Overall, the results show that the primary efficiency advantage of LL-E, RLL-E, and fs-RLL-E comes from avoiding backbone replication. The price paid for function space repulsion is increased training-time computation and memory consumption, while inference remains close to that of a single model.

Table 5: Feature space dimension of different base network architectures.

| | |
|---|---|
| RESNET-18 | $d = 512$ |
| WIDERESNET-28-10 | $d = 640$ |
| RESNET-50 | $d = 2048$ |
| VIT-B/16 | $d = 768$ |

Table 6: **Representative efficiency profiling with ResNet18.** Overheads are reported relative to the single ResNet18 model. For DE-10, peak memory is reported per independently trained member.

| Method | Members | Params (M) | Inference overhead | Training overhead | Peak memory (MB) |
|---|---|---|---|---|---|
| Single | 1 | 11.23 | $1.00\times$ | $1.00\times$ | 534 |
| LL-E | 10 | 11.69 | $1.01\times$ | $1.08\times$ | 541 |
| RLL-E | 10 | 11.69 | $1.01\times$ | $1.21\times$ | 541 |
| fs-RLL-E | 10 | 11.69 | $1.01\times$ | $2.07\times$ | 883 |
| DE-10 | 10 | 112.28 | $9.74\times$ | $9.84\times$ | 534 |
| RDE-3 | 3 | 33.68 | $2.96\times$ | $5.85\times$ | 2474 |

---

[2]A microbatch is a subset of the repulsion batch processed in one forward/backward pass. With gradient accumulation, several microbatches are processed sequentially and their gradients are accumulated before a single optimizer update, yielding the same effective batch size with lower peak memory usage.

Table 7: **Representative efficiency profiling with ResNet50.** Overheads are reported relative to the single ResNet50 model. For DE-10, peak memory is reported per independently trained member.

| Method | Members | Params (M) | Inference overhead | Training overhead | Peak memory (MB) |
|--------|---------|-----------|-------------------|-------------------|------------------|
| Single | 1 | 23.71 | $1.00\times$ | $1.00\times$ | 1767 |
| LL-E | 10 | 25.58 | $1.01\times$ | $1.01\times$ | 1795 |
| RLL-E | 10 | 25.58 | $1.02\times$ | $1.11\times$ | 1795 |
| fs-RLL-E | 10 | 25.58 | $1.02\times$ | $2.03\times$ | 3115 |
| DE-10 | 10 | 237.15 | $10.01\times$ | $10.08\times$ | 1767 |
| RDE-3 | 3 | 71.14 | $3.04\times$ | $5.92\times$ | 8908 |

Table 8: **Absolute latency measurements.** Inference latency is reported per image; training-step latency is reported per optimization step. Values are mean $\pm$ std in milliseconds.

| Backbone | Method | Inference / image (ms) | Train step (ms) |
|----------|--------|------------------------|-----------------|
| ResNet18 | Single | $0.407 \pm 0.004$ | $23.29 \pm 0.14$ |
| ResNet18 | LL-E | $0.410 \pm 0.003$ | $25.26 \pm 0.29$ |
| ResNet18 | RLL-E | $0.413 \pm 0.003$ | $28.25 \pm 0.66$ |
| ResNet18 | fs-RLL-E | $0.413 \pm 0.002$ | $48.28 \pm 0.33$ |
| ResNet18 | DE-10 | $3.968 \pm 0.014$ | $229.16 \pm 0.46$ |
| ResNet18 | RDE-3 | $1.207 \pm 0.016$ | $136.19 \pm 0.34$ |
| ResNet50 | Single | $1.276 \pm 0.016$ | $66.62 \pm 0.10$ |
| ResNet50 | LL-E | $1.286 \pm 0.014$ | $67.39 \pm 0.12$ |
| ResNet50 | RLL-E | $1.306 \pm 0.010$ | $73.98 \pm 0.26$ |
| ResNet50 | fs-RLL-E | $1.306 \pm 0.014$ | $135.19 \pm 0.26$ |
| ResNet50 | DE-10 | $12.774 \pm 0.031$ | $671.49 \pm 5.94$ |
| ResNet50 | RDE-3 | $3.885 \pm 0.026$ | $394.22 \pm 0.70$ |

Table 9: **Memory-runtime trade-off for fs-RLL-E with gradient accumulation.** All configurations use the same effective in-distribution and repulsion batch sizes of 16. Training overheads are relative to the corresponding single model.

| Backbone | Microbatch / accumulation | Training overhead | Peak memory (MB) |
|----------|---------------------------|-------------------|------------------|
| ResNet18 | 16 / 1 | $2.07\times$ | 883 |
| ResNet18 | 8 / 2 | $3.97\times$ | 409 |
| ResNet18 | 4 / 4 | $9.34\times$ | 328 |
| ResNet50 | 16 / 1 | $2.03\times$ | 3115 |
| ResNet50 | 8 / 2 | $2.29\times$ | 1222 |
| ResNet50 | 4 / 4 | $4.94\times$ | 887 |

## A.2 Datasets

*DirtyMNIST:* This dataset (Mukhoti et al., 2023) consists of 60,000 clean MNIST digits with unique class labels and 60,000 synthetically generated ambiguous digits with multiple labels. All images are grayscale and of resolution 28×28.

*CIFAR10 / CIFAR100:* The original datasets (Krizhevsky et al., 2009) contain 50,000 training and 10,000 test images. CIFAR-10 has 10 classes; CIFAR-100 has 100 fine-grained classes. All images are 32×32 RGB.

*CIFAR10-C / CIFAR100-C:* These benchmarks (Hendrycks & Dietterich, 2019) contain corrupted versions of the CIFAR-10 and CIFAR-100 test sets. Each consists of 10,000 test images per corruption type across 19 corruption types and 5 severity levels. Image resolution is 32×32 RGB.

*Fine-tuning datasets:* For high-resolution evaluation, we use Stanford Cars (Krause et al., 2013) and Food101 (Bossard et al., 2014). Stanford Cars contains 16,185 images (8,144 train / 8,041 test) and 196 classes. Food101 consists of 101,000 images and 101 classes (750 train / 250 test per class). All inputs are center-cropped and resized to 224×224.

*OOD benchmarks:* For OOD detection on DirtyMNIST, we use FashionMNIST (Xiao et al., 2017), eMNIST (Cohen et al., 2017), and Omniglot (Lake et al., 2015) as OOD datasets. For the remaining datasets, we use Places365 (Zhou et al., 2017), SVHN (Netzer et al., 2011), ImageNet-O (Hendrycks et al., 2021), Country211 (Radford et al., 2021), and TinyImageNet (Le & Yang, 2015).

*Repulsion samples:* For the repulsion samples, we use kMNIST (Clanuwat et al., 2018), Textures (Cimpoi et al., 2014), and ImageNet-21k (Deng et al., 2009).

# B Additional experiments

## B.1 Uncertainty decomposition for active learning

We evaluate the performance of our fs-RLL-E uncertainty estimates on an active learning task proposed in (Mukhoti et al., 2023). Given a small number of initial training samples and a large pool of unlabeled data, the goal is to iteratively select the most informative samples to improve model performance.

We initialize training with 20 labeled samples and use a pool composed of clean and ambiguous MNIST samples in a 1:60 ratio. In each acquisition step, the 5 samples with the highest uncertainty are selected and added to the training set. Since ambiguous samples are inherently noisy, disentangling aleatoric and epistemic uncertainty is essential for avoiding uninformative acquisitions. After each acquisition, the network is retrained from scratch on the extended dataset.

As shown in Fig. 5, acquisition based on predictive entropy (PE), both for single models and deep ensembles, consistently performs worse. These methods tend to select ambiguous samples with high aleatoric uncertainty, which are of limited value for improving model performance. In contrast, acquisition strategies based on epistemic uncertainty – such as DEs, DDU, LL-E, and fs-RLL-E – are more effective in avoiding these uninformative samples.

However, despite this qualitative advantage, all epistemic acquisition strategies perform comparably to the random baseline. This finding is consistent with recent results by Gashi et al. (Gashi et al., 2025), who show that many active learning methods fail to outperform simple baselines across a range of experimental settings.

At the same time, our results highlight the limitations of predictive entropy as an acquisition function in the presence of significant aleatoric uncertainty. While epistemic uncertainty-based methods may not yield substantial improvements in final accuracy under the current setup, they do provide more principled guidance in avoiding harmful acquisitions.

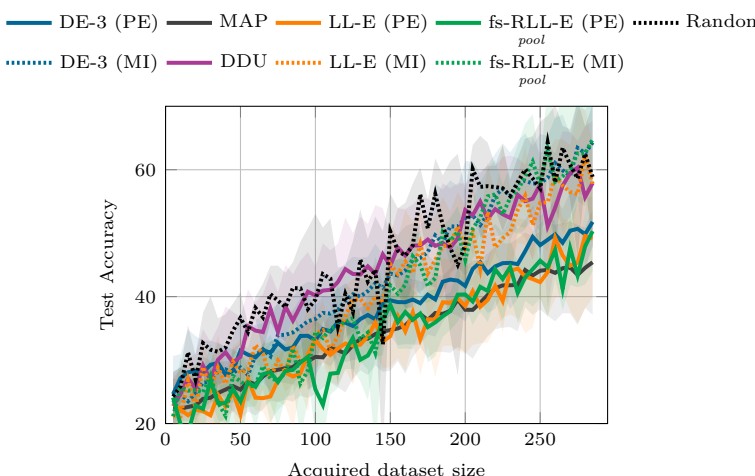

Figure 5: Test accuracy of the model as a function of the data samples that are acquired using the different uncertainty estimates. Predictive entropy (PE) combines aleatoric and epistemic uncertainty. Using the mutual information (MI) of the LL-E and fs-RLL-E prediction outperforms softmax entropy of the single network and performs on par with the other uncertainty baselines. The results are averaged over 5 runs.

## B.2  Repulsion sample ablations in transfer learning

We complement the transfer-learning results with an analysis of repulsion sample selection. We aim to answer the question of whether any distribution outside the training data is sufficient for function space repulsion, or whether the repulsion samples must be structured and informative. We therefore conduct two ablation experiments. First, Tables 10 and 11 compare repulsion samples from different natural datasets for Food101 and Stanford Cars, respectively, while reporting both ID performance and average OOD detection. Second, Table 12 studies whether the benefit of ImageNet-21k repulsion samples is retained when samples from a diverse set are interpolated towards either ID images or unstructured noise.

Tables 10 and 11 show that changing the repulsion dataset has only a small effect on ID accuracy and NLL, but a much larger effect on OOD detection. Repulsion on ID images is consistently weaker than repulsion on structured auxiliary OOD data. On Food101, ImageNet-21k improves average OOD AUROC from 87.40% for LL-E to 97.31% for ResNet18 and from 96.40% to 98.66% for ResNet50. On Stanford Cars, the same pattern is visible: ImageNet-21k improves average OOD AUROC from 93.18% to 99.35% for ResNet18 and from 96.37% to 99.34% for ResNet50. Textures also performs strongly, especially compared to ID repulsion, which confirms that the gain is not simply caused by adding any repulsion loss.

Table 12 further tests whether the structure of the auxiliary samples matters. Starting from ImageNet-21k repulsion samples for fs-RLL-E, we interpolate them either toward Food101 ID images or toward random noise. In both cases, OOD detection decreases as the interpolation weight increases. This shows that the benefit of ImageNet-21k is gradually lost when the repulsion samples become either too close to the ID distribution or too unstructured. In particular, moving fully toward Food101 ID images recovers the weaker ID-repulsion behavior from Table 10, while moving toward random noise approaches the performance of an uninformative repulsion distribution.

Overall, these ablations support the main practical message: structured auxiliary OOD samples provide the most reliable improvements for function space repulsion given a shared representation. Repulsion on ID images is comparatively weak and can harm OOD detection, while replacing auxiliary OOD samples with noise degrades performance toward the unregularized LL-E baseline. At the same time, the ID metrics remain comparatively stable, suggesting that the main effect of repulsion sample selection concerns uncertainty quality rather than raw predictive accuracy.

Table 10: **Food101: Repulsion-sample ablation with joint ID and OOD metrics.** For each backbone, we report in-distribution accuracy [%] / NLL and average OOD AUROC [%] / AUPR [%] across OOD benchmarks. Results are averaged over 5 seeds. Best OOD results for each backbone are **bold**, second-best are underlined.

| Method | Uncertainty | Repulsion | ResNet18 | | ResNet50 | |
| | | | ID | OOD | ID | OOD |
| | | | Acc. / NLL | AUROC / AUPR | Acc. / NLL | AUROC / AUPR |
| --- | --- | --- | --- | --- | --- | --- |
| | Entropy | | | 95.65 / 91.85 | | 92.66 / 82.77 |
| MAP | MSP | – | 75.73 / 0.94 | 93.55 / 83.70 | 82.44 / 0.72 | 90.04 / 76.93 |
| | Max Logit | | | 71.28 / 37.53 | | 38.86 / 22.36 |
| LL-E | MI | – | 75.71 / 0.94 | 87.40 / 65.54 | 82.22 / 0.72 | 96.40 / 92.50 |
| fs-RLL-E (ours) | MI | Food101 (ID) | 75.81 / 0.93 | 86.38 / 63.70 | 82.50 / 0.72 | 93.62 / 82.69 |
| fs-RLL-E (ours) | MI | Stanford Cars | 75.86 / 0.93 | 91.16 / 77.81 | 82.54 / 0.72 | 96.91 / 93.13 |
| fs-RLL-E (ours) | MI | Flowers102 | 75.90 / 0.93 | 89.21 / 70.29 | 82.48 / 0.72 | 95.59 / 87.34 |
| fs-RLL-E (ours) | MI | Textures | 75.89 / 0.93 | 95.28 / 88.90 | 82.31 / 0.72 | 97.61 / 94.73 |
| fs-RLL-E (ours) | MI | ImageNet-21k | 75.79 / 0.94 | **97.31 / 94.73** | 82.61 / 0.72 | **98.66 / 97.36** |

Table 11: **Stanford Cars: Repulsion-sample ablation with joint ID and OOD metrics.** For each backbone, we report in-distribution accuracy [%] / NLL and average OOD AUROC [%] / AUPR [%] across OOD benchmarks. Results are averaged over 5 seeds. Best OOD results for each backbone are **bold**, second-best are underlined.

| Method | Uncertainty | Repulsion | ResNet18 | | ResNet50 | |
| | | | ID | OOD | ID | OOD |
| | | | Acc. / NLL | AUROC / AUPR | Acc. / NLL | AUROC / AUPR |
| --- | --- | --- | --- | --- | --- | --- |
| | Entropy | | | 90.94 / 91.63 | | 96.00 / 96.66 |
| MAP | MSP | – | 66.39 / 1.25 | 86.99 / 84.77 | 71.91 / 1.07 | 92.14 / 92.64 |
| | Max Logit | | | 68.24 / 63.89 | | 12.86 / 37.28 |
| LL-E | MI | – | 68.22 / 1.17 | 93.18 / 93.70 | 71.98 / 1.06 | 96.37 / 96.62 |
| fs-RLL-E (ours) | MI | Stanford Cars (ID) | 68.26 / 1.18 | 92.67 / 93.20 | 71.63 / 1.07 | 95.05 / 95.38 |
| fs-RLL-E (ours) | MI | Food101 | 68.26 / 1.17 | 97.00 / 97.58 | 71.86 / 1.06 | 97.33 / 97.52 |
| fs-RLL-E (ours) | MI | Flowers102 | 68.59 / 1.16 | 97.07 / 97.68 | 71.83 / 1.06 | 96.98 / 97.43 |
| fs-RLL-E (ours) | MI | Textures | 68.19 / 1.18 | 99.21 / 99.43 | 71.98 / 1.06 | 98.85 / 99.13 |
| fs-RLL-E (ours) | MI | ImageNet-21k | 68.22 / 1.17 | **99.35 / 99.54** | 71.70 / 1.07 | **99.34 / 99.49** |

Table 12: **Food101: Repulsion-sample composition ablation.** Average OOD AUROC [%]. Results are averaged over 5 seeds. At 0%, repulsion uses only ImageNet-21k samples; larger weights move the samples toward either random noise or Food101 ID images.

| Target distribution | 0% | 25% | 50% | 75% | 100% |
| --- | --- | --- | --- | --- | --- |
| Noise | 97.31 | 94.19 | 91.55 | 89.34 | 89.32 |
| Food101 ID | 97.31 | 96.10 | 90.93 | 87.88 | 86.38 |

### B.3 Full ensembling versus pretraining with post-hoc uncertainty

**Setup** We evaluate epistemic uncertainty estimates on OOD detection tasks using CIFAR-10 and CIFAR-100 as training datasets. We train a ResNet50 from scratch with spectral normalization to mitigate feature collapse. For the pretrained models, we use a ResNet50 and ViT-B/16 pretrained on ImageNet-1k, using the weight checkpoint provided by Torchvision.

For the base model trained from scratch, we reinitialize the final linear layer and retrain an ensemble of 10 linear heads, keeping the backbone frozen. We use the AdamW optimizer with a learning rate of $10^{-4}$. For the ImageNet-1k pretrained backbones, we follow the same protocol but fine-tune the heads and backbone jointly. We use a learning rate of $10^{-3}$ for the linear ensemble heads and $10^{-5}$ for the backbone. To obtain the single-model (MAP) baseline for the ImageNet-1k pretrained model, we use one linear layer and the same training procedure as for the last-layer ensembles. For function space repulsion, we use minibatches from the Textures dataset as repulsion samples. As OOD datasets we use SVHN (Netzer et al., 2011), Places365 (Zhou et al., 2017), ImageNet-O (Hendrycks et al., 2021), Country211 (Radford et al., 2021), and TinyImageNet (Le & Yang, 2015).

To ensure a fair comparison between models trained from scratch and ImageNet-pretrained models, we matched the ID/OOD preprocessing steps for each scenario: the low-resolution models were trained and evaluated at native CIFAR-10/100 resolution ($32 \times 32$), with OOD data accordingly downsampled. Conversely, pretrained models fine-tuned on CIFAR-10/100 at $224 \times 224$ were evaluated on OOD samples first downsampled to CIFAR resolution and then re-upscaled to avoid access to more detailed OOD data.

It has previously been shown that pretrained models can improve uncertainty estimates of a single model (Hendrycks et al., 2019a). However, as demonstrated in the DirtyMNIST experiment (see Section 6.2), a single model is not sufficient to disentangle aleatoric and epistemic uncertainty. While this may not be evident for high-performing models, it leads to performance degradation when in-distribution data exhibits increased aleatoric uncertainty due to label ambiguity or a lack of prediction accuracy.

When using a pretrained model as a fixed checkpoint, deep ensembles lose the diversity from random weight initialization. Moreover, full deep ensembles are resource-heavy (parameters/time), making post-hoc and last-layer approaches attractive when one wants to reduce backbone replication. The central question is therefore not whether these methods uniformly outperform full deep ensembles, but whether they improve uncertainty estimates while maintaining strong predictive performance. In this experiment, we aim to answer the following questions:

(i) Is a last-layer ensemble sufficiently diverse for epistemic uncertainty estimation? Can repulsion in parameter or function space recover missing diversity?

(ii) Given the choice, should we train a full deep ensemble from scratch, or use a pretrained model extended with post-hoc uncertainty methods?

**Results** Tables 14 and 15 summarize the ID performance and OOD detection under semantic shift, respectively. For the ID performance, we report accuracy and NLL scores. Since our main objective is to estimate epistemic uncertainty, we do not primarily expect the repulsion terms to improve accuracy or NLL. Instead, we view these metrics as quantities that should be preserved while uncertainty estimates improve. We use OOD detection as a proxy task for evaluating epistemic uncertainty, following the assumption that uncertainty should increase for unfamiliar inputs (de Mathelin et al., 2025).

We begin by evaluating a ResNet50 model trained from scratch with random initialization. Across all post-hoc uncertainty methods (LL-Laplace, LL-E, RLL-E, and fs-RLL-E), ID performance remains comparable to the single-model (MAP) baseline. In contrast, the deep ensemble (DE-5) improves accuracy by 1 to 3%. This indicates that full deep ensembles remain a strong baseline when predictive accuracy is the main objective. This performance gain is consistent with prior work suggesting that ensembles benefit from learning more diverse features through different initializations (Allen-Zhu & Li, 2023).

Next, we consider a ResNet50 model pretrained on ImageNet-1k. Fine-tuning this model on CIFAR-10 and CIFAR-100 yields strong accuracy and calibration, matching DE-5 on CIFAR100 and outperforming it on

Table 13: **Model configurations and parameter counts.** We report the number of trainable parameters for each base model and the increase introduced by a LL-E with 10 particles. DE-5 replicates the entire base model. LL-E shares the backbone and only replicates the final classifier layer. Reported values reflect configurations used in Tables 14 and 15 for CIFAR-10 and CIFAR-100.

| Base model | Input Res. | # Parameters | |
| --- | --- | --- | --- |
| | | 10 classes | 100 classes |
| ResNet50 | 32×32 | 20.74M | 20.93M |
| × 5 (DE-5) | | 103.72M | 104.64M |
| + (fs)-RLL-E | | + 0.20M | + 2.05M |
| ResNet50 | 224×224 | 23.53M | 23.71M |
| + (fs)-RLL-E | | + 0.20M | + 2.05M |
| ViT-B/16 | 224×224 | 85.81M | 85.88M |
| + (fs)-RLL-E | | + 77k | + 0.77M |

CIFAR10. Using a more powerful backbone such as ViT-B/16 further improves ID metrics and results in the best overall performance. Much of the predictive-performance gain in this comparison stem from pretraining. The uncertainty methods are evaluated by the additional uncertainty improvements they provide on top of the base model. Table 13 reports the parameter count for each model configuration.

We now turn to OOD detection based on epistemic uncertainty. For models trained from scratch, DE-5 performs best on CIFAR10 with an AUROC of 91.15%, followed by DDU with 90.90%. In this setting, our methods do not outperform these baselines. However, on CIFAR100, function space repulsion improves AUROC from 79.83% for LL-E to 84.05% for fs-RLL-E, outperforming the other post-hoc methods. Thus, fs-RLL-E is not uniformly best, but it can improve uncertainty estimates while keeping in-distribution performance comparable.

The benefit of pretraining becomes more apparent in OOD detection. On ResNet50 and ViT-B/16, even the entropy-based uncertainty of a single model (MAP) outperforms DE-5. For example, MAP with softmax entropy on pretrained ResNet50 achieves an AUROC and AUPR of 94.19 and 93.36% on CIFAR10. Adding a LL-E improves these scores to 94.85 and 94.36%, while fs-RLL-E reaches 94.95 and 95.10%. On CIFAR100, the AUROC increases from 82.29% with MAP entropy to 84.62% with fs-RLL-E on ResNet50, and from 86.22% to 87.87% on ViT-B/16.

DDU, which was the best-performing OOD detector on DirtyMNIST, performs less effectively on this benchmark. The increased dimensionality of the feature space and the larger number of classes in CIFAR100 limit the applicability of the full GMM model. The diagonal covariance approximation and the assumption of class-conditional Gaussian features may be too restrictive. Similarly, the max logit baseline performs well on ViT-B/16 but degrades notably on ResNet-based models. LL-Laplace, while less effective on DirtyMNIST, achieves the best results in this setting.

In summary, using pretrained models not only improves ID classification performance but also creates a strong basis for epistemic uncertainty estimation. A single pretrained model achieves similar or better results than DE-5 in several settings, and repulsion in function space with fs-RLL-E provides additional improvements over the unregularized LL-E. At the same time, LL-Laplace remains a strong baseline, and DE or DDU may be preferable in some regimes.

We note that CIFAR10 and CIFAR100 consist of low-resolution images, which may limit clear separability between ID and OOD samples. In Section 6.4, we considered fine-grained image classification tasks with high-resolution images. There, we demonstrated the applicability of our fs-RLL-E, especially using a more diverse set of repulsion samples.

Table 14: **In-distribution performance (Accuracy [%] / Negative Log Likelihood (NLL)).** We compare base models that are either trained from *random initialization* or *fine-tuned from ImageNet-1k pretrained weights.* Results are averaged over 5 seeds.

| Method | CIFAR10 | | | CIFAR100 | | |
|---|---|---|---|---|---|---|
| | **ResNet50** (Random) | **ResNet50** (Pretrained) | **ViT-B/16** (Pretrained) | **ResNet50** (Random) | **ResNet50** (Pretrained) | **ViT-B/16** (Pretrained) |
| MAP | 94.93 / 0.1992 | 96.39 / 0.1171 | 98.29 / 0.0757 | 79.74 / 0.8087 | 81.86 / 0.6852 | 88.16 / 0.5604 |
| LL-Laplace | 94.93 / 0.1886 | 96.24 / 0.1180 | 98.27 / 0.0641 | 79.67 / 0.7873 | 80.83 / 0.7501 | 86.94 / 0.6191 |
| LL-E | 94.67 / 0.2005 | 96.49 / 0.1138 | 98.02 / 0.0797 | 79.00 / 0.8133 | 81.99 / 0.6722 | 88.17 / 0.5671 |
| RLL-E *(ours)* | 94.66 / 0.2005 | 96.44 / 0.1151 | 98.26 / 0.0705 | 79.00 / 0.8133 | 81.98 / 0.6768 | 88.28 / 0.5574 |
| fs-RLL-E *(ours)* + TEXTURES | 94.77 / 0.1923 | 96.38 / 0.1214 | 98.30 / 0.0732 | 78.69 / 0.8227 | 81.90 / 0.6809 | 87.09 / 0.5870 |
| DE-5 | 96.10 / 0.1247 | — / — | — / — | 82.88 / 0.6270 | — / — | — / — |

Table 15: **Semantic shift detection (AUROC [%] / AUPR [%]).** We compare base models that are either trained from *random initialization* or *fine-tuned from ImageNet-1k pretrained weights.* Results are averaged over 5 seeds. Best results for each base model are **bold**, second-best are underlined.

| Method | Uncertainty | CIFAR10 | | | CIFAR100 | | |
|---|---|---|---|---|---|---|---|
| | | **ResNet50** (Random) | **ResNet50** (Pretrained) | **ViT-B/16** (Pretrained) | **ResNet50** (Random) | **ResNet50** (Pretrained) | **ViT-B/16** (Pretrained) |
| MAP | MSP | 88.71 / 86.12 | 93.45 / 91.63 | 97.07 / 96.91 | 79.54 / 76.74 | 80.33 / 77.83 | 85.09 / 83.23 |
| | Max Logit | 88.97 / 87.93 | 93.08 / 90.07 | 97.92 / 98.34 | 80.13 / 77.23 | 75.28 / 71.68 | 89.32 / 89.24 |
| | Entropy | 89.04 / 87.24 | 94.19 / 93.36 | 97.3 / 97.49 | 80.29 / 77.36 | 82.29 / 80.25 | 86.22 / 85.7 |
| LL-Laplace | MI | 89.22 / 87.01 | **95.97** / **96.11** | **97.99** / **98.48** | 81.27 / 78.43 | **86.35** / **85.32** | **89.91** / **89.16** |
| DDU (diag) | GMM density | 90.90 / 87.54 | 87.4 / 88.54 | 94.90 / 94.41 | 80.26 / 76.92 | 72.09 / 71.44 | 87.19 / 85.01 |
| LL-E | MI | 88.91 / 86.9 | 94.85 / 94.36 | 97.43 / 97.92 | 79.83 / 77.37 | 83.19 / 81.85 | 87.45 / 86.88 |
| RLL-E *(ours)* | MI | 89.15 / 87.76 | 94.02 / 92.48 | 97.69 / 98.15 | 80.45 / 78.24 | 83.57 / 80.7 | 87.92 / 87.75 |
| fs-RLL-E *(ours)* + TEXTURES | MI | 88.36 / 86.52 | 94.95 / 95.10 | 97.78 / 98.33 | **84.05** / **81.0** | 84.62 / 83.81 | 87.87 / 88.15 |
| DE-5 | MI | **91.15** / **87.24** | — / — | — / — | 79.04 / 75.28 | — / — | — / — |

