# OpenReview forum: "Function Space Diversity for Uncertainty Prediction via Repulsive Last-Layer Ensembles"
_TMLR — Under review for TMLR_

### Review · Reviewer_ESoZ · 2025-11-24

**Summary Of Contributions:**

This work aims at closing the gap between deep ensampbes (DEs) and particle-optimisation variational inference (POVI) performance, by introducing a method that exploits last-layers ensembles (LL-E) with a repulsion mechanism that compensate the low diversity.

This paper is a bit out of my expertise, nonetheless, I find it clear and well-written. It provides good insights, and the claims are backed up by a solid experimental section. The only possible weakness might be the slightly incremental contribution, but this is well balanced by the quality of the text, and the good (experimental and theoretical) insights throughout the entire paper.

Just two very minor typos/mistakes in the backgrounds:
- third line, K is not defined (easy to guess but still)
- above eq. 1, \epsilon_l is defined but doesn't appear in any equation

**Audience:**

Yes

**Audience Explanation:**

Good insights are provided that will for sure be of interest to the TMLR's audience.

**Claims And Evidence:**

Yes

**Claims Explanation:**

The experimental section is quite extensive and back-ups all the claims made by the authors.

**Requested Changes:**

Nothing.

---

> ### Author Response · Authors · 2026-06-12
>
> We thank the reviewer for the positive feedback and for pointing out these notation issues. We will define $K$ explicitly and fix the inconsistency around $\epsilon_l$ in the revised manuscript.

---

### Review · Reviewer_o8bx · 2025-11-27

**Summary Of Contributions:**

The paper proposes repulsive last-layer ensembles (RLL-E and fs-RLL-E), which apply particle-optimization variational inference in function space to a multi-headed network sharing a backbone, giving a parameter-efficient way to approximate function-space posteriors and to add uncertainty heads on top of pretrained models.  The authors argue that previous function-space POVI methods underperform mainly due to practical choices (independent full networks and repulsion on training data) and instead enforce diversity on unlabeled OOD or label-destroyed inputs, which better shapes epistemic uncertainty and OOD behavior.   They provide experiments on synthetic regression/classification, DirtyMNIST, CIFAR10/100-C, and transfer learning with ImageNet-pretrained ResNets and ViT showing that RLL-E/fs-RLL-E often match or surpass DE-5, LL-Laplace, LL-E, and diagonal DDU in OOD detection and calibration while leaving ID accuracy essentially unchanged.

**Audience:**

Yes

**Audience Explanation:**

The work sits squarely at the intersection of Bayesian deep learning, uncertainty quantification, and practical deployment with pretrained models. The paper advances function-space inference and repulsive ensembles, two active research directions, by providing a more scalable architecture and concrete guidance on using auxiliary OOD data or augmentations to shape epistemic uncertainty. The empirical study over DirtyMNIST, CIFAR-C, and transfer-learning benchmarks with modern architectures (ResNet, ViT) is likely to interest practitioners who want lightweight uncertainty methods for real systems without training full ensembles from scratch.

**Broader Impact Concerns:**

None.

**Claims And Evidence:**

Yes

**Claims Explanation:**

* The central claim that *"a multi-head last-layer parameterization is sufficient for function-space POVI and yields diverse predictions"* is clearly illustrated on synthetic regression/classification in the toy examples (Section 6.1, Figure 3).

* The claim that *"the method better disentangles aleatoric and epistemic uncertainty and improves OOD detection"* is supported by DirtyMNIST results, where RLL-E/fs-RLL-E achieve higher AUROC/AUPR than MAP, LL-Laplace, LL-E, and DE-5 on both clean-vs-OOD and ambiguous-vs-OOD tasks (Section 6.2, Table 1).

* The robustness and calibration claims under covariate shift are backed by CIFAR10-C/100-C experiments, where fs-RLL-E improves NLL and matches or improves ECE relative to deep ensembles across corruption severities (Section 6.3, Figure 4).

* Some aspects, including generality beyond vision tasks and sensitivity to the choice of repulsion data, are only partially explored, but the authors acknowledge and discuss these issues, so the evidence for the main claims is overall convincing if not exhaustive (Sections 6.2, 6.4, and the discussion around repulsion data).

**Requested Changes:**

Overall, the authors have done a commendable job, and the following minor suggestions may further help readers deepen their understanding:

* Provide a more systematic analysis of how the choice of repulsion samples affects performance (e.g., varying how far they are from the ID data and showing failure cases), building on the existing DirtyMNIST and semantic OOD experiments.

* Briefly discuss the limitations of mutual-information-based epistemic uncertainty and why MI is still used here, in the uncertainty/related-work or discussion sections.

* Add a short “practical guidance” paragraph summarizing when to prefer fs-RLL-E/RLL-E over DDU, LL-Laplace, or standard LL ensembles, based on patterns in the main result tables.

---

> ### Author Response · Authors · 2026-06-12
>
> We thank the reviewer for the positive assessment and constructive suggestions. We address the individual points in detail below.
>
> ### (1) Repulsion-sample selection and failure modes
> > **Provide a more systematic analysis of how the choice of repulsion samples affects performance (e.g., varying how far they are from the ID data and showing failure cases), building on the existing DirtyMNIST and semantic OOD experiments.**
>
> We address this point in the shared response on repulsion-sample selection above. The added ablations vary the repulsion distribution of auxiliary OOD samples, ID samples, and noise-only samples. These experiments test how the distance and structure of the repulsion samples affect performance and illustrate failure cases where repulsion is either too close to the ID manifold or too far from semantically relevant regions.
>
> ### (2) Limitations of MI-based epistemic uncertainty
> > **Briefly discuss the limitations of mutual-information-based epistemic uncertainty and why MI is still used here, in the uncertainty/related-work or discussion sections.**
>
> We use mutual information because it is a standard ensemble-based disagreement measure and has been widely used in prior work on Bayesian neural networks, deep ensembles, and uncertainty decomposition. In the revised manuscript, we will clarify that we use MI operationally as a measure of predictive disagreement between ensemble members. This is useful for the downstream tasks considered in the paper, such as OOD detection and separating ambiguous from epistemically uncertain inputs. Importantly, we will also mention recent work questioning the interpretation of entropy-based uncertainty decompositions and state this as a limitation of our evaluation.
>
> ### (3) Practical guidance
> > **Add a short “practical guidance” paragraph summarizing when to prefer fs-RLL-E/RLL-E over DDU, LL-Laplace, or standard LL ensembles, based on patterns in the main result tables.**
>
> We agree that this makes the paper more useful for practitioners. **In the revision, we add a practical guidance paragraph summarizing when each method is preferable based on the observed empirical patterns.**
>
> In brief, we recommend *LL-E* as a simple and cheap baseline when one wants to add multiple prediction heads to a shared backbone. *RLL-E* is useful when additional diversity is desired but no reliable auxiliary repulsion samples are available. *fs-RLL-E* is preferable when meaningful auxiliary OOD data or label-destroying augmentations are available, especially when OOD detection or epistemic uncertainty under shift is the target.
>
> Compared to *LL-Laplace*, fs-RLL-E can explicitly incorporate knowledge about task-relevant OOD regions through the choice of repulsion samples. LL-Laplace remains a strong last-layer Bayesian baseline, but it is tied to a local posterior approximation around the trained solution and does not provide the same mechanism for enforcing disagreement on selected input regions. *DDU* can perform strongly when the latent feature space satisfies its class-conditional density assumptions, but can degrade when these assumptions are violated or when covariance approximations are insufficient.
>
> *Full deep ensembles* remain a strong option when compute and memory are not limiting. However, in transfer-learning settings, a single pretrained backbone can already match or outperform deep ensembles trained from scratch. **Using repulsive uncertainty heads can combine strong pretrained representations with improved uncertainty estimates without training several full backbones.**

---

### Review · Reviewer_L3YQ · 2026-06-09

**Summary Of Contributions:**

This paper proposes a function-space repulsive last-layer ensemble for uncertainty estimation. The method shares a feature extractor across ensemble members and applies repulsion only to the last-layer heads, aiming to obtain useful epistemic uncertainty at lower cost than full deep ensembles.

### **Strengths**
- The method is simple, parameter-efficient, and well-motivated.
- A key empirical finding is that the choice of repulsion samples is critical: enforcing repulsion on training samples can be harmful, while OOD samples or label-destroying augmentations can improve uncertainty estimates.

### **Weaknesses**
- The narrow focus on vision classification, which does not fully support the broader claims about general uncertainty-aware fine-tuning.
- The limited evidence for claims about large pretrained networks
- The paper focuses on uncertainty, but predictive accuracy should be treated as a necessary constraint. The accuracy–uncertainty trade-off is not sufficiently discussed, especially where full deep ensembles remain stronger.
- The efficiency claim is under-supported. Parameter count alone is insufficient; training time, inference cost, memory usage, and other comparisons are needed.

**Audience:**

Yes

**Audience Explanation:**

The paper should interest researchers working on uncertainty quantification, Bayesian deep learning, OOD detection, calibration, and efficient ensembles. The paper addresses an important and practical problem: obtaining reliable uncertainty estimates without the high cost of full deep ensembles.

**Claims And Evidence:**

No

**Claims Explanation:**

Partially. The experiments support the core claim that function-space repulsion can improve last-layer ensemble uncertainty when suitable repulsion samples are used.

However, several broader claims are not fully supported:
- The evaluation is almost entirely limited to standard vision classification benchmarks with relatively relatively small-to-moderate-size backbones, which does not fully support the broader claims about large pretrained networks or general uncertainty-aware fine-tuning.
- Some uncertainty improvements appear to come with reduced accuracy
- The method depends strongly on the choice of repulsion samples, but the paper provides limited guidance on when particular repulsion distributions should work or fail.
- The efficiency claim is also incomplete, since parameter count alone does not establish practical savings without training time, inference cost, memory usage, or compute-normalized comparisons. In addition,
- The exposition is sometimes unclear, with insufficiently explained terminology and weak narrative organization.

**Requested Changes:**

- Support the claims about large pretrained networks and general uncertainty-aware fine-tuning with stronger evidence beyond vision classification and moderate-size models.
- Discuss the accuracy–uncertainty trade-off more explicitly, especially in settings where full deep ensembles remain stronger.
- Add quantitative efficiency comparisons, including training time, inference cost, memory usage, and compute-normalized results.
- Provide clearer guidance and failure-mode analysis for repulsion-sample selection.
- The presentation could be substantially improved.

---

> ### Author Response · Authors · 2026-06-12
>
> We thank the reviewer for the constructive feedback. We address the comments point by point below.
>
> ### (1) Scope of claims and evidence
>
> > **Support the claims about large pretrained networks and general uncertainty-aware fine-tuning with stronger evidence beyond vision classification and moderate-size models.**
>
> We thank the reviewer for raising this point. With the phrase “large pretrained networks,” we intended to refer to pretrained backbones that are computationally relevant in the context of Bayesian neural networks and particle-based inference, where jointly optimizing several ResNet or ViT backbones can be demanding. However, we now recognize that this wording can be ambiguous, especially in the broader context of foundation models. We therefore revise the wording and refer more precisely to “pretrained vision backbones.”
>
> We will also clarify the empirical scope of the paper. The proposed formulation, a shared representation with multiple diverse prediction heads, is general in principle, but our experiments validate it on vision classification tasks. **In the revision, we make clear that the evidence supports pretrained vision backbones rather than arbitrary modalities, foundation models, or downstream tasks.** Regression, language, segmentation, and larger foundation models are stated as limitations and future work.
>
> ### (2) Accuracy-uncertainty trade-off
> > **Discuss the accuracy–uncertainty trade-off more explicitly, especially in settings where full deep ensembles remain stronger.**
>
> We agree that predictive accuracy should be treated as a necessary constraint. The primary goal of RLL-E/fs-RLL-E is not to improve ID accuracy, but to improve uncertainty estimates while preserving predictive performance. This is why the original submission reports ID accuracy and NLL alongside uncertainty metrics. **In the revision, we will make this interpretation more explicit and discuss cases where uncertainty improvements come with reduced in-domain performance as accuracy-uncertainty trade-offs rather than unconditional improvements.**
>
> We also clarify the role of pretrained backbones in this comparison. Full deep ensembles trained from scratch often improve accuracy over a single model trained from scratch and provide uncertainty estimates through diversity across independently trained models. Pretrained backbones, however, can also provide strong ID accuracy. Adding repulsive last-layer heads can then combine the predictive strength of pretrained representations with additional uncertainty estimates, without training several full backbones.

---

> > ### Author Response · Authors · 2026-06-12
> >
> > ### (3) Efficiency and computational overhead
> > > **Add quantitative efficiency comparisons, including training time, inference cost, memory usage, and compute-normalized results.**
> >
> > We agree that parameter count alone does not fully characterize efficiency. We therefore separate the discussion into parameter storage, inference overhead, training overhead, and GPU memory, and add representative empirical profiling results below.
> >
> > **Parameter count / storage.**
> >
> > The main efficiency benefit of RLL-E/fs-RLL-E is backbone sharing. A full deep ensemble stores $M$ complete DNN models. In contrast, LL-E, RLL-E, and fs-RLL-E store one backbone and $M$ last-layer heads. Thus, **the parameter count increases over a single DNN only through the additional classifier heads**, rather than by replicating the full backbone.
> >
> > **Inference overhead.**
> >
> > At inference time, the repulsion losses are not evaluated. The backbone features are computed once, and the ensemble predictions are obtained by applying $M$ linear classifier heads. For a feature vector of dimension $d$ and $C$ classes, this corresponds to replacing one classifier matrix in $\mathbb{R}^{C \times d}$ by one larger linear projection in $\mathbb{R}^{MC \times d}$. **Therefore, LL-E, RLL-E, and fs-RLL-E have the same inference graph and near-single-model inference latency**, while DE-$M$ requires $M$ full backbone evaluations.
> >
> > **Training overhead.**
> >
> > Compared to a single DNN, LL-E adds only the cost of evaluating $M$ last-layer heads on the same backbone features, which is small relative to the backbone forward/backward pass. RLL-E adds a repulsion term on the last-layer parameters. fs-RLL-E adds the main additional training cost: an extra repulsion batch and pairwise prediction distances between the $M$ heads on this batch. The pairwise computation scales as $O(M^2N_{\mathrm{rep}}C)$, where $N_{\mathrm{rep}}$ is the repulsion batch size and $C$ is the number of classes. Unlike DE-$M$ and RDE-$M$, fs-RLL-E avoids training multiple complete backbones. While DE members can be trained independently, RDE-$M$ requires joint optimization because its repulsion objective couples the complete models.
> >
> > **GPU memory.**
> >
> > Training memory includes parameters, optimizer states, gradients, activations, and temporary tensors. LL-E and RLL-E primarily add the cost of the additional classifier heads and multi-head predictions while retaining a single shared backbone. fs-RLL-E additionally requires a forward and backward pass on the repulsion batch. When the backbone is fine-tuned, this repulsion loss also backpropagates through the shared backbone.
> > Peak memory can be reduced using smaller microbatches and gradient accumulation while maintaining the same effective ID and repulsion batch sizes. This trades memory for additional training time. RDE requires more peak memory because it jointly trains several complete backbone models.
> >
> > **Empirical quantification.**
> >
> > We profile a single DNN, LL-E, RLL-E, fs-RLL-E, independently trained DE-10, and jointly trained RDE-3 using ResNet18 and ResNet50 on the same NVIDIA RTX A4000. All last-layer ensemble methods use $M=10$, and the profiling benchmark uses an effective ID and repulsion batch size of 16. We report inference latency, effective training-step latency, parameter count, and peak training memory in Table E1-E4.
> >
> > For DE-10, the reported latency includes sequentially processing all ten members on one GPU, while peak training memory is reported per independently trained member. RDE-3 is trained jointly because its repulsion objective couples the complete models.
> >
> > The results show that LL-E, RLL-E, and fs-RLL-E retain near-single-model inference latency because they share the backbone. Standard fs-RLL-E approximately doubles training-step latency due to the additional repulsion pass. Smaller repulsion microbatches and gradient accumulation can reduce peak memory at the cost of increased training time (see Table E4). DE-10 and RDE-3 require substantially more total computation because they evaluate or optimize multiple complete backbones.
> >
> > Accordingly, we revise our claim: **RLL-E and fs-RLL-E reduce backbone replication, parameter count, and inference cost relative to full deep ensembles. fs-RLL-E introduces an explicit training-time and memory overhead whose magnitude depends on the repulsion batch size and memory-management strategy.**

---

> > > ### Author Response · Authors · 2026-06-12
> > > **Empirical quantification: Results**
> > >
> > > **Table E1: Efficiency profiling with ResNet18.** Overheads are relative to the single ResNet18 model. For DE-10, peak memory is per independently trained member.
> > >
> > > | Method   |   Members |   Params (M) | Inference time overhead   | Training time overhead   |   Peak memory (MB) |
> > > |:---------|----------:|-------------:|:--------------------------|:-------------------------|-------------------:|
> > > | Single   |         1 |        11.23 | 1.00×                     | 1.00×                    |                534 |
> > > | LL-E     |        10 |        11.69 | 1.01×                     | 1.08×                    |                541 |
> > > | RLL-E    |        10 |        11.69 | 1.01×                     | 1.21×                    |                541 |
> > > | fs-RLL-E |        10 |        11.69 | 1.01×                     | 2.07×                    |                883 |
> > > | DE-10    |        10 |       112.28 | 9.74×                     | 9.84×                    |                534 |
> > > | RDE-3    |         3 |        33.68 | 2.96×                     | 5.85×                    |               2474 |
> > >
> > >
> > > **Table E2: Efficiency profiling with ResNet50.** Overheads are relative to the single ResNet50 model. For DE-10, peak memory is per independently trained member.
> > >
> > > | Method   |   Members |   Params (M) | Inference time overhead   | Training time overhead   |   Peak memory (MB) |
> > > |:---------|----------:|-------------:|:--------------------------|:-------------------------|-------------------:|
> > > | Single   |         1 |        23.71 | 1.00×                     | 1.00×                    |               1767 |
> > > | LL-E     |        10 |        25.58 | 1.01×                     | 1.01×                    |               1795 |
> > > | RLL-E    |        10 |        25.58 | 1.02×                     | 1.11×                    |               1795 |
> > > | fs-RLL-E |        10 |        25.58 | 1.02×                     | 2.03×                    |               3115 |
> > > | DE-10    |        10 |       237.15 | 10.01×                    | 10.08×                   |               1767 |
> > > | RDE-3    |         3 |        71.14 | 3.04×                     | 5.92×                    |               8908 |
> > >
> > >
> > > **Table E3: Absolute latency measurements.** Inference latency is reported per image; training-step latency is reported per optimization step. Values are mean ± std in milliseconds.
> > >
> > > | Backbone   | Method   | Inference / image (ms)   | Train step (ms)   |
> > > |:-----------|:---------|:-------------------------|:------------------|
> > > | ResNet18   | Single   | 0.407 ± 0.004            | 23.29 ± 0.14      |
> > > | ResNet18   | LL-E     | 0.410 ± 0.003            | 25.26 ± 0.29      |
> > > | ResNet18   | RLL-E    | 0.413 ± 0.003            | 28.25 ± 0.66      |
> > > | ResNet18   | fs-RLL-E | 0.413 ± 0.002            | 48.28 ± 0.33      |
> > > | ResNet18   | DE-10    | 3.968 ± 0.014            | 229.16 ± 0.46     |
> > > | ResNet18   | RDE-3    | 1.207 ± 0.016            | 136.19 ± 0.34     |
> > > | ResNet50   | Single   | 1.276 ± 0.016            | 66.62 ± 0.10      |
> > > | ResNet50   | LL-E     | 1.286 ± 0.014            | 67.39 ± 0.12      |
> > > | ResNet50   | RLL-E    | 1.306 ± 0.010            | 73.98 ± 0.26      |
> > > | ResNet50   | fs-RLL-E | 1.306 ± 0.014            | 135.19 ± 0.26     |
> > > | ResNet50   | DE-10    | 12.774 ± 0.031           | 671.49 ± 5.94     |
> > > | ResNet50   | RDE-3    | 3.885 ± 0.026            | 394.22 ± 0.70     |
> > >
> > >
> > > **Table E4: Memory–runtime tradeoff for fs-RLL-E with gradient accumulation.** All configurations use the same effective ID and repulsion batch sizes of 16. Training overheads are relative to the corresponding single model.
> > >
> > > | Backbone   | Microbatch    |   Accumulation steps | Training overhead   |   Peak memory (MB) |
> > > |:-----------|:--------------|---------------------:|:--------------------|-------------------:|
> > > | ResNet18   | 16            |                    1 | 2.07×               |                883 |
> > > | ResNet18   | 8             |                    2 | 3.97×               |                409 |
> > > | ResNet18   | 4             |                    4 | 9.34×               |                328 |
> > > | ResNet50   | 16            |                    1 | 2.03×               |               3115 |
> > > | ResNet50   | 8             |                    2 | 2.29×               |               1222 |
> > > | ResNet50   | 4             |                    4 | 4.94×               |                887 |

---

> > > > ### Author Response · Authors · 2026-06-12
> > > >
> > > > ### (4) Repulsion-sample selection and failure modes
> > > > > **Provide clearer guidance and failure-mode analysis for repulsion-sample selection.**
> > > >
> > > > We address this point in the shared response on repulsion-sample selection above. In brief, we add a practitioner-oriented guide, an ablation over different repulsion datasets in the pretrained transfer-learning setting, and an additional mixture/noise ablation. Together, these results clarify when ID repulsion, auxiliary OOD samples, and noise-only samples are helpful or harmful, and provide concrete guidance for choosing repulsion samples in practice.
> > > >
> > > > ### (5) Presentation
> > > > > **The presentation could be substantially improved.**
> > > >
> > > > We will improve the presentation by adding clearer transitions in our motivation, method, and experiments, providing missing definitions, and adding a practical guidance paragraph summarizing when each method is preferable (cf. response (3) to reviewer o8bx).

---

### Author Response · Authors · 2026-06-12
**General Comment**

We thank the reviewers for their thoughtful and constructive feedback.

In accordance with the TMLR review process, we use this response as a discussion framework for the revision. Below, we summarize four issues that were identified as the main aspects, the changes we make, and the additional analyses we include. We then address the individual reviewer comments in more detail. A revised version of the paper will be uploaded in the coming days.

1. **Issue:** *Focus on vision classification.*

    **Mitigation:** We clarify the scope of our claims throughout the paper. The framework is general in the sense that the repulsive ensemble is defined on top of a shared representation, but the empirical validation in this paper is limited to vision classification tasks. We therefore revise statements about “large pretrained networks” and “general uncertainty-aware fine-tuning” to make clear that the paper provides evidence for pretrained vision backbones, including ResNet and ViT models, rather than for arbitrary modalities, foundation models, or downstream tasks.

2. **Issue:** *Predictive accuracy.*

     **Mitigation:** We make predictive accuracy an explicit constraint when interpreting uncertainty improvements. The goal of RLL-E/fs-RLL-E is not to improve in-distribution accuracy over full deep ensembles, but to improve uncertainty estimates while preserving predictive performance and avoiding full backbone replication. We therefore discuss OOD detection, calibration, ID accuracy, and NLL jointly, and explicitly highlight cases where uncertainty improvements come with accuracy or NLL trade-offs.

3. **Issue:** *Repulsion sample selection.*

     **Mitigation:** We expand the discussion of repulsion-sample selection and add a practitioner-oriented explanation of when repulsion samples are expected to help or fail. Further, we add ablation experiments over different repulsion datasets in the pretrained transfer-learning setting, as well as an ablation replacing auxiliary OOD samples with ID images or noise *(see Tables R1a-R3 in the shared response below)*.

4. **Issue:** *Efficiency evaluation.*

     **Mitigation:** We clarify the efficiency discussion by distinguishing between parameter count, inference time overhead, training time overhead, and memory usage. The key practical advantage of RLL-E/fs-RLL-E is reduced backbone replication: at inference time, the shared-backbone ensemble requires one backbone forward pass and multiple lightweight head evaluations, whereas a full ensemble requires multiple full backbone evaluations. During training, fs-RLL-E incurs additional costs due to the repulsion batch and pairwise prediction distances. We state this explicitly, add representative profiling results, and avoid generalized claims such as “minimal compute and memory.”

These changes will make the paper’s claims more precise, clarify the accuracy–uncertainty trade-off, and provide more actionable guidance on when the proposed method should be preferred over LL-E, LL-Laplace, DDU, or full deep ensembles.

---

> ### Author Response · Authors · 2026-06-12
> **Shared response: Repulsion-sample selection and failure modes**
>
> Reviewers *L3YQ* and *o8bx* both commented on the role of repulsion samples. We therefore address this central point here.
>
> The main question is where diversity should be enforced. The existing DirtyMNIST experiment illustrates one failure case: repulsion on the training distribution itself does not provide useful epistemic diversity for separating ambiguous ID samples from OOD samples. We extend this analysis to the transfer-learning setting with pretrained models.
>
> We analyze whether any distribution beyond ID data is sufficient for repulsion, or whether the repulsion samples should be structured natural auxiliary data. We add two ablations:
>  1. We vary the repulsion datasets: ID data, and auxiliary OOD datasets
>  2. Starting from auxiliary OOD repulsion samples, we move towards the training data distribution and towards random noise.
>
> Our results show that structured auxiliary OOD datasets provide the most reliable improvements. Repulsion on ID data harms OOD detection, even when ID accuracy remains stable. Repulsion on random noise does not recover the benefits of auxiliary OOD samples and behaves much closer to LL-E.
>
>
> ### **Ablation over repulsion datasets (1/2).**
> First, we performed an ablation in the pretrained transfer-learning setting where we vary the repulsion samples for Food101 and StanfordCars (for ResNet18 and ResNet50). Tables R1a and R2a show that ID accuracy and NLL are mostly stable across repulsion choices. **In contrast, Tables R1b and R2b show that OOD performance depends strongly on where diversity is enforced.**
> Across these settings, repulsion on ID samples preserves ID performance but can degrade OOD detection compared to LL-E. Auxiliary OOD samples, especially diverse datasets such as ImageNet21k or Textures, provide much more consistent OOD improvements.
>
> **Table R1a: ID performance on Food101.**
>
> | Method | Repulsion samples | ResNet18 — Accuracy [%] / NLL | ResNet50 — Accuracy [%] / NLL |
> | --- | --- | --- | --- |
> | Single | - | 75.73 / 0.94 | 82.44 / 0.72 |
> | LL-E | - | 75.71 / 0.94 | 82.22 / 0.72 |
> | fs-RLL-E | Food101 (ID) | 75.81 / 0.93 | 82.50 / 0.72 |
> | fs-RLL-E | StanfordCars | 75.86 / 0.93 | 82.54 / 0.72 |
> | fs-RLL-E | Flowers102 | 75.90 / 0.93 | 82.48 / 0.72 |
> | fs-RLL-E | Textures | 75.89 / 0.93 | 82.31 / 0.72 |
> | fs-RLL-E | ImageNet21k | 75.79 / 0.94 | 82.61 / 0.72 |
>
> **Table R1b: Average OOD performance on Food101.**
>
> | Method | Uncertainty | Repulsion samples | ResNet18 — Average AUROC [%] / AUPR [%] | ResNet50 — Average AUROC [%] / AUPR [%] |
> | --- | --- | --- | --- | --- |
> | Single | Entropy | - | 95.65 / 91.85 | 92.66 / 82.77 |
> | Single | MaxLogit | - | 71.28 / 37.53 | 38.86 / 22.36 |
> | LL-E | MI | - | 87.40 / 65.54 | 96.40 / 92.50 |
> | fs-RLL-E | MI | Food101 (ID) |  86.38 / 63.70  | 93.62 / 82.69 |
> | fs-RLL-E | MI | StanfordCars | 91.16 / 77.81 | 96.91 / 93.13 |
> | fs-RLL-E | MI | Flowers102 | 89.21 / 70.29 | 95.59 / 87.34 |
> | fs-RLL-E | MI | Textures | 95.28 / 88.90 | 97.61 / 94.73 |
> | fs-RLL-E | MI | ImageNet21k | **97.31** / **94.73** | **98.66** / **97.36** |
>
> **Table R2a: ID performance on StanfordCars.**
>
> | Method | Repulsion samples | ResNet18 — Accuracy [%] / NLL | ResNet50 — Accuracy [%] / NLL |
> | --- | --- | --- | --- |
> | Single | - | 66.39 / 1.25 | 71.91 / 1.07 |
> | LL-E | - | 68.22 / 1.17 | 71.98 / 1.06 |
> | fs-RLL-E | StanfordCars (ID) | 68.26 / 1.18 | 71.63 / 1.07 |
> | fs-RLL-E | Food101 | 68.26 / 1.17 | 71.86 / 1.06 |
> | fs-RLL-E | Flowers102 | 68.59 / 1.16 | 71.83 / 1.06 |
> | fs-RLL-E | Textures | 68.19 / 1.18 | 71.98 / 1.06 |
> | fs-RLL-E | ImageNet21k | 68.22 / 1.17 | 71.70 / 1.07 |
>
> **Table R2b: Average OOD performance on StanfordCars.**
>
> | Method | Uncertainty | Repulsion samples | ResNet18 — Average AUROC [%] / AUPR [%] | ResNet50 — Average AUROC [%] / AUPR [%] |
> | --- | --- | --- | --- | --- |
> | Single | Entropy | - | 90.94 / 91.63 | 96.00 / 96.66 |
> | Single | MaxLogit | - | 68.24 / 63.89 | 12.86 / 37.28 |
> | LL-E | MI | - | 93.18 / 93.70 | 96.37 / 96.62 |
> | fs-RLL-E | MI | StanfordCars (ID) | 92.67 / 93.20 | 95.05 / 95.38 |
> | fs-RLL-E | MI | Food101 | 97.00 / 97.58 | 97.33 / 97.52 |
> | fs-RLL-E | MI | Flowers102 | 97.07 / 97.68 | 96.98 / 97.43 |
> | fs-RLL-E | MI | Textures | 99.21 / 99.43 | 98.85 / 99.13 |
> | fs-RLL-E | MI | ImageNet21k | **99.35** / **99.54** | **99.34** / **99.49** |

---

> ### Author Response · Authors · 2026-06-12
>
> ### **Ablation over repulsion datasets (2/2).**
>
> We analyze the effect of moving from auxiliary OOD repulsion samples to ID samples or to unstructured noise. We use Food101 as the ID training dataset, ImageNet21k as the auxiliary OOD dataset, and uniform pixel noise in the normalized input space. Starting from the OOD samples, we construct repulsion samples by interpolating them either with ID samples or with noise. We conduct two interpolation sweeps
>
> $$
> x_{\mathrm{rep}} = (1-w)x_{\mathrm{aux}} + wx_{\mathrm{ID}}
> $$
>
> and
>
> $$
> x_{\mathrm{rep}} = (1-w)x_{\mathrm{aux}} + wx_{\mathrm{noise}}.
> $$
>
> Each cell in Table R3 reports average OOD AUROC for the indicated mixture weight $w$. At $w=0\%$, repulsion uses only ImageNet21k samples. As $w$ increases, the repulsion samples move toward Food101 ID images or unstructured noise.
>
> **Table R3: Average OOD AUROC [%] for the repulsion-sample composition ablation on Food101.**
>
> | Target distribution / mixture weight | 0% | 25% | 50% | 75% | 100% |
> |:---|:---:|:---:|:---:|:---:|:---:|
> | Noise | **97.31** | 94.19 | 91.55 | 89.34 | 89.32 |
> | Food101 ID | **97.31** | 96.10 | 90.93 | 87.88 | 86.38  |
>
>
> Our ablations show that OOD detection is strongest when using structured auxiliary OOD repulsion samples. Replacing them with Food101 ID images substantially degrades OOD detection, showing that **ID repulsion can hurt** by enforcing disagreement near the training manifold. Replacing auxiliary samples with unstructured noise also reduces performance. **Noise appears too far from the relevant data manifold to induce useful epistemic uncertainty**, leading to behavior closer to unregularized LL-E. Food101 ID accuracy remains approximately constant at 75.7–75.9%, indicating that the effect mainly concerns OOD uncertainty rather than ID prediction.
>
>
> ### **Practical guidance.**
>
> This leads to the following practical guidance. **When OOD detection is the target and suitable auxiliary data is available, auxiliary natural OOD samples are the preferred choice for function-space repulsion.** ID samples should not be the default repulsion distribution, as they can harm OOD performance. If no auxiliary OOD data is available, label-destroying augmentations can be considered, but their effectiveness is dataset-dependent and should be validated empirically. Noise-only samples are far from the ID distribution, but may be too unstructured to induce useful epistemic disagreement and do not recover the gains of structured auxiliary OOD data.

---

### Author Response · Authors · 2026-06-22
**Revision**

We have uploaded a revised version of our paper, incorporating the changes discussed in our response to the reviewers. Specifically, we

* *clarified the scope and positioning of the method*, emphasizing that our empirical evaluation focuses on pretrained vision backbones for image-classification tasks;
* *clarified that predictive accuracy is treated as a constraint* when interpreting improvements in uncertainty estimation;
* *added ablation studies on the choice of repulsion samples* and repeated the transfer-learning experiments in the main text with additional random seeds (10 seeds instead of 5);
* *added a practitioner's guide* and expanded the limitations and future-work discussion;
* *added transition sentences* to improve the flow and readability of the paper;
* *clarified the efficiency discussion* by distinguishing between parameter count, inference-time overhead, training-time overhead, and memory usage, and by adding profiling experiments.

We hope that the revised manuscript addresses the reviewers' concerns and clarifies the practical contribution and scope of the proposed method. We thank the reviewers again for their constructive feedback.